# LEARNING TO ANTICIPATE: A CONDITIONAL REPRESENTATION FUSION NETWORK FOR PRE-STROKE PREDICTION

## ABSTRACT

Predicting the future in dynamic environments requires reasoning about the intentions of agents from rich, multi-modal data. We introduce a novel machine learning problem: pre-intervention anticipation—forecasting outcomes before an action is completed by fusing contextual cues with ongoing sensor data. To address this, we propose ConFu, a general neural architecture featuring two key innovations: (1) a conditional gating mechanism that dynamically modulates primary features (e.g., trajectory) based on secondary context (e.g., intention cues), and (2) a cross-fusion strategy for systematic multi-stage integration of heterogeneous modalities. **Our model achieves a prediction accuracy of 92.6% with a mean absolute error of 0.20 meters, significantly outperforming existing methods by 7.8-10.5% in accuracy**. Experimental validation on a real-world badminton dataset comprising 13,582 strokes demonstrates that ConFu provides immediate tactical feedback, saving 85% decision time compared to trajectory-based approaches. This time advantage is particularly valuable for practical applications such as enabling badminton robots to compute interception strategies.

Our work establishes a foundation for intention-aware prediction, with broader implications for robotics, autonomous systems, and human-AI interaction. Code will be released for reproducibility (**https://anonymous.4open.science/r/AI-Sport18-BFE9/README.md** (needed you to paste it into browser by yourself) and supplementary material by now).

## 1 INTRODUCTION

The sports analytics market is projected to grow at a compound annual rate of 21.3% from 2021 to 2028 Research (2021), driven by the convergence of artificial intelligence and sports science. This paradigm shift enables data-driven insights into athletic performance, strategy formulation, and training optimization Davis et al. (2024). While traditional systems like Hawk-Eye provide high-precision tracking using multi-camera setups Uzor et al. (2023); Singh Bal & Dureja (2012), their deployment cost and infrastructure requirements limit scalability. In contrast, monocular vision-based deep learning methods offer a low-cost, accessible alternative, capable of extracting rich spatio-temporal signals for predictive modeling.

Among racket sports, badminton presents a particularly challenging domain for real-time anticipation due to its rapid rally dynamics—players often have less than 500ms between consecutive strokes Wolf Gawin & Seidler (2015). Prior work has focused on match outcome prediction Sharma et al. (2021) or post-hoc statistical analysis of stroke sequences Torres-Luque et al. (2020; 2019), which offer limited utility for in-the-moment decision-making. More recent efforts analyze player positioning Galeano et al. (2021) and stroke patterns via Markov models Galeano et al. (2022), highlighting the importance of fine-grained movement understanding. However, these approaches typically operate *after* stroke execution, failing to support proactive responses.

To enable truly anticipatory systems, we advocate for **pre-stroke prediction**: forecasting where a shuttlecock will land using only observations available *before* or *at the instant of* impact. This novel problem represents a fundamental shift from inferring the landing point from the ball's trace to predicting it from pre-stroke contextual cues, enabling significantly shorter inference time and making

it particularly suitable for robotic interception planning where rapid decision-making is crucial. This requires fusing multiple modalities—such as the evolving 3D trajectory of the shuttlecock, player body pose, arm gestures, and inferred stroke intent—into a coherent, time-critical prediction. The core challenge lies in dynamically integrating a primary sensory stream (e.g., shuttlecock motion) with contextual cues (e.g., player gesture) that modulate its interpretation. Naïve fusion strategies struggle to capture these conditional dependencies, especially under the tight temporal constraints of elite play.

To address this, we propose **ConFu** (Conditional Gated Cross-Fusion Network), a novel architecture for multimodal pre-stroke anticipation in badminton. ConFu unifies four key information streams from monocular video: (1) reconstructed 3D shuttlecock trajectories, (2) player dynamic localization, (3) keypoint-based arm gestures, and (4) predicted stroke types. By leveraging conditional gating and hierarchical cross-fusion, our model generates accurate predictions of the shuttlecock's landing location precisely at the moment of opponent contact. We evaluate ConFu on real-world datasets TrackNetV2 Sun et al. (2020) and ShuttleSet22 Wang et al. (2024b), demonstrating significant improvements over baseline methods. The primary contributions of this research are as follows:

1. **Real-time Prediction Capability**: ConFu achieves drop point prediction within 0.224 seconds after stroke initiation, enabling 85% time saving compared to post-stroke trajectory methods;

2. **Comprehensive Multimodal Integration**: We systematically combine four information modalities (3D trajectory, player positioning, arm gestures, and stroke types) extracted from monocular video, achieving 92.6% prediction accuracy;

3. **Novel Gating Mechanisms**: We design two specialized conditional gating mechanisms—dynamic spatio-temporal fusion and stroke-conditioned gesture filtering—that improve prediction accuracy by 3.3-10.5% over baseline fusion strategies;

4. **Hierarchical Cross-Fusion Architecture**: The proposed cross-fusion approach integrates features across multiple processing stages, preserving original information while enabling deep feature interaction.

Beyond immediate drop point forecasting, our approach lays the foundation for intelligent training systems, wearable feedback interfaces, and autonomous badminton-playing robots capable of human-level reactivity. By bridging multimodal perception with anticipatory reasoning, ConFu represents a step toward real-time, intention-driven sports intelligence.

## 2 RELATED WORK

**3D Trajectory Prediction.** Early approaches primarily relied on physics-based models to analyze shuttlecock trajectories Yi et al. (2004); Chen et al. (2009); Zhang et al. (2010), though these required precise calibration. Recent deep learning methods Sato et al. (2024); Chao et al. (2024) have improved tracking in high-speed scenarios using event cameras and sequence models. The TrackNet series Huang et al. (2019); Sun et al. (2020); Chen & Wang (2024) and MonoTrack Liu & Wang (2022) demonstrate the potential of 3D trajectory reconstruction from monocular video, enabling cost-effective deployment.

**Multimodal Fusion for Prediction.** Existing studies have explored using laser scanners Waghmare et al. (2016), trajectory points Vrajesh et al. (2020), and player posture Wu et al. (2019); Wu & Koike (2020) for landing point prediction. Shimizu et al. Shimizu et al. (2019) combined position and posture information to predict tennis shot direction, while Chang et al. Chang et al. (2023) proposed DyMF, a dynamic graph model for player action prediction. Compared to existing works like ShuttleNet Wang et al. (2021), our approach differs in: (1) utilizing richer multimodal inputs extracted directly from pre-stroke video frames, and (2) introducing a cross-fusion mechanism that enables deep interaction between modalities while preserving their specific characteristics.

**Fusion Strategies.** Multimodal fusion is typically categorized as early, middle, or late fusion Baltrusaitis et al. (2019). Early fusion Barnum et al. (2020) combines raw features but may introduce noise, late fusion Snoek et al. (2005) processes modalities independently but overlooks cross-modal dependencies, while middle fusion Wang et al. (2024a) enables interaction at intermediate stages.

Our cross-stage fusion strategy facilitates hierarchical feature interaction through attention mechanisms, preventing the loss of critical information during deep fusion processes.

**Datasets.** High-quality badminton datasets include BadmintonDB Ban et al. (2022), the ShuttleSet series Wang et al. (2023; 2024b), and the TrackNet series. This study utilizes TrackNetV2 and ShuttleSet22, comprising 33,612 strokes, to support fine-grained performance evaluation.

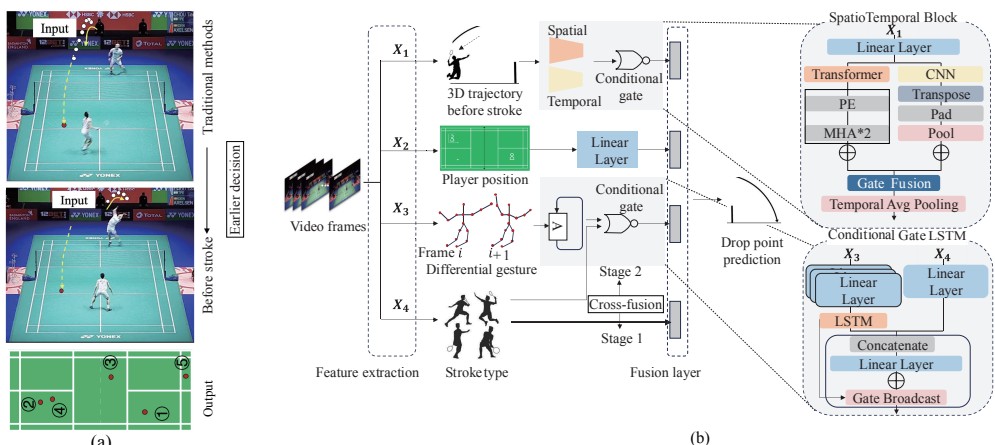

Figure 1: Illustration of the ConFu architecture. The model includes four inputs: the 3D trajectory of shuttlecock before stroke ($X_1$), two players' positions ($X_2$), the gesture feature ($X_3$) and stroke type ($X_4$), with conditional gates put on $X_1$ and $X_3$ to enable dynamic feature recalibration, and outputs the shuttlecock drop point prediction.

# 3 METHODOLOGY

## 3.1 ARCHITECTURE OVERVIEW

ConFu is designed to address the dependency on post-stroke trajectories of badminton drop point prediction, enabling accurate and instantaneous predictions at the moment of stroke. As illustrated in Figure 1, the system integrates four input modalities: (1) 3D shuttlecock trajectory reconstructed from monocular video, (2) spatio-temporal player coordinates, (3) gesture features from arm keypoints, and (4) stroke types. The spatial-temporal fusion is driven by the need to model the two fundamental components of a **3D trajectory**: spatial geometry and temporal dynamics. To this end, we employ dedicated architectures—a CNN for spatial structure and a Transformer for temporal dependencies. Since the importance of spatial geometry and temporal dynamics varies depending on the specific context, we use conditional fusion allows the model to dynamically determine which domain to prioritize for each input. We propose a unified framework that combines a conditional gating mechanism with a cross-fusion architecture to effectively integrate multimodal features for shuttlecock drop point prediction. The conditional gate dynamically models inter-modality interactions by computing gating values via a sigmoid function, allowing the model to adaptively modulate auxiliary information. Meanwhile, the cross-fusion architecture systematically integrates features across multiple processing stages, preserving the original characteristics of each modality while enabling comprehensive information fusion.

The core novelty of our framework lies in addressing the new and challenging problem of pre-stroke landing prediction, which fundamentally differs from traditional approaches that rely on post-stroke trajectory analysis. Our design introduces two key innovations: (1) dynamic spatio-temporal decomposition that extracts features from different domains (spatial and temporal) and merges them dynamically, and (2) conditional cross-stage fusion that dynamically prioritizes modalities based on the real-time inferred stroke context.

## 3.2 CROSS-FUSION MECHANISM

To enable deep interaction while preserving original modality representations, we design a hierarchical cross-fusion module that operates across two stages.

Let $F^{(1)} = [H_1, H_2, H_3, H_4]$ denote the modality-specific features from Stage 1, and $F^{(2)} = [H_1', H_2', H_3', H_4']$ from Stage 2. The cross-fusion is defined as:

$$F_{\text{fused}} = \text{Concat}\left(F^{(1)}, \text{CrossAttn}(F^{(1)}, F^{(2)})\right) \tag{1}$$

where $\text{CrossAttn}(Q, K, V) = \text{Softmax}\left(\frac{QK^T}{\sqrt{d}}\right)V$, with $Q = F^{(1)}W_Q$, $K = F^{(2)}W_K$, $V = F^{(2)}W_V$. This allows Stage 2 features to modulate Stage 1 representations via attention, preserving early features while enabling late-stage refinement.

This allows Stage 2 features to modulate Stage 1 representations via attention, preserving early features while enabling late-stage refinement.

Unlike standard fusion approaches that simply concatenate all modalities at the final stage, our multi-stage fusion strategy incorporates intermediate fusion to capture intrinsic relationships between different data streams. Ablation studies show that this approach contributes complementary benefits over common used fusion strategy including spatio-temporal fusion, conditional gating, and cross-fusion, together outperforming one-shot fusion.

### 3.3 SPATIO-TEMPORAL TRAJECTORY ENCODER

The 3D trajectory of the shuttlecock is reconstructed via MonoTrack Liu & Wang (2022) from monocular video of badminton match. To capture the non-linear dynamics of shuttlecock motion, we employ a dual-branch transformer architecture that separately models temporal dependencies and spatial relationships. The input $X_1 \in \mathbb{R}^{3 \times T}$ (3D coordinates over $T = 21$ frames) is processed as follows:

**Spatio-Temporal Transformation** The temporal branch uses a two-layer transformer encoder with multi-head self-attention (short name: MHA, we use 2 heads in our experiment) to model long-range dependencies across frames, while the spatial branch applies local windowed attention (window size=3) to capture instantaneous velocity/acceleration patterns:

$$\mathbf{H}_1^{(t)} = \text{TemporalTransformer}(W_1 X_1 + B_1), \tag{2}$$

$$\mathbf{H}_1^{(s)} = \text{SpatialTransformer}(W_1 X_1 + B_1), \tag{3}$$

where the weight matrix $W_1 \in \mathbb{R}^{d \times 3}$ and the bias matrix $B_1 \in \mathbb{R}^{d \times T}$ perform affine transformations to map coordinates into a $d$-dimensional hidden space.

**Dynamic Fusion** A learnable gating mechanism adaptively balances temporal and spatial features:

$$g_1 = \sigma(W_{g_1}[\mathbf{H}_1^{(t)} \oplus \mathbf{H}_1^{(s)}] + b_{g_1}), \tag{4}$$

$$\mathbf{H}_1 = \mu(g_1 \odot \mathbf{H}_1^{(t)} + (1 - g_1) \odot \mathbf{H}_1^{(s)}), \tag{5}$$

where $\sigma(\cdot)$ and $\mu(\cdot)$ denote the Sigmoid function and temporal averaging over the sequence, respectively, and $W_{g_1} \in \mathbb{R}^{d \times 2d}$ generates fusion weights. The final trajectory feature $\mathbf{H}_1$ is obtained by averaging the fused sequence over time.

### 3.4 CONDITIONAL GATE LSTM

To model stroke preparation dynamics, we extract $T$ frames of 2D pixel coordinates of six key points on the player's arms before the stroke, followed by first-order differencing to obtain the gesture feature $X_3 \in \mathbb{R}^{12 \times (T-1)}$. Then the feature is analyzed using a LSTM conditioned on stroke type $X_4$:

$$\mathbf{h} = \text{LSTM}(W_3 X_3 + b_3), \quad W_3 \in \mathbb{R}^{d \times 12}, \tag{6}$$

where $\mathbf{h} = (\mathbf{h}_1, \mathbf{h}_2, \ldots, \mathbf{h}_{T-1})' \in \mathbb{R}^{(T-1) \times d}$ includes the $d$-dimensional hidden states of totally $T - 1$ timesteps. A stroke-conditional gate filters irrelevant motion patterns:

$$g_3 = \sigma(W_{g_3}[\mathbf{h}_{T-1} \oplus \mathbf{H}_4] + b_{g_3}), \tag{7}$$

$$\mathbf{H}_3 = \mathbf{h} \odot g_3, \tag{8}$$

where $W_{g_3} \in \mathbb{R}^{d \times 2d}$ and $\mathbf{H}_4 = \text{Embed}(X_4) \in \mathbb{R}^d$ is the stroke type embedding. This mechanism suppresses noise from non-stroke-related movements (e.g., footwork adjustments). The integration of explicit stroke labels ($X_4$) enables the model to learn discriminative gesture features for different shot types (smash, drive, etc.)

Although Transformer architectures have demonstrated advantages in modeling long-range temporal dependencies, we conducted an ablation study comparing LSTM and Transformer for gesture encoding. While Transformer achieves comparable accuracy (92.1% vs 92.6%), it incurs significantly higher inference time (158ms vs 97ms). Given our emphasis on real-time responsiveness and model transparency, the conditional LSTM is better suited for short-sequence, pre-stroke gesture prediction.

**Formalization as Contextual Modulation.** We formalize the conditional gating mechanism as a *contextual feature modulation* operation. Let $H_1 \in \mathbb{R}^d$ be the trajectory feature vector and $H_3 \in \mathbb{R}^d$ be the gesture feature vector. The gating network $G(\cdot)$ maps $H_3$ to a scale $\gamma$ and shift $\beta$ vector:

$$(\gamma, \beta) = G(H_3) = (\sigma(W_\gamma H_3 + b_\gamma),\ W_\beta H_3 + b_\beta) \tag{9}$$

The modulated feature $\tilde{H}_1$ is then:

$$\tilde{H}_1 = \gamma \odot H_1 + \beta \tag{10}$$

This is analogous to Conditional Batch Normalization, where the gesture $H_3$ provides the conditioning context. This formulation allows the model to not only scale but also shift the trajectory features based on intention, enabling richer interaction.

**Interpretability via Gate Weights.** The gating weight $w_g = \sigma(\text{MLP}(H_3))$ can be interpreted as an *intention-driven attention map*. In Figure 3, we visualize $w_g$ for different stroke types. We observe that for a *smash*, the gate assigns higher weights to the latter part of the trajectory (near the stroke), as the player's intention dominates. For a *clear*, the gate assigns more uniform weights, indicating reliance on the full trajectory. This provides insight into the model's decision-making process.

The ground-truth labels for the shuttlecock's landing point ($y \in \mathbb{R}^2$) are generated by integrating the 3D trajectory reconstructed by MonoTrack with a calibrated aerodynamic model. This process is crucial for supervised learning and is therefore detailed here to ensure reproducibility and address potential concerns regarding label quality and feature-label coupling.

During our implementation, we identified that the default aerodynamic damping parameters in the public MonoTrack codebase often produced physically implausible trajectories, likely due to a miscalibration. This manifested as trajectories that were excessively shortened, failing to align with the visual evidence in the video frames. Furthermore, its method of approximating the landing point by simply interpolating frames where the shuttlecock's height ($z$-coordinate) changes sign introduces significant error. To ensure the highest label fidelity for training and evaluation, we meticulously re-calibrated the physical model and implemented a more precise landing point calculation.

The motion of a shuttlecock in flight is governed by gravity and aerodynamic drag. Its dynamics can be modeled by the following equation of motion:

$$m\frac{d^2\vec{r}}{dt^2} = m\vec{g} - \frac{1}{2}C_d\rho A\|\vec{v}\|\vec{v} \tag{11}$$

where $\vec{r}$ is the position vector, $m$ is the mass of the shuttlecock, $\vec{g}$ is the gravitational acceleration vector, $C_d$ is the drag coefficient, $\rho$ is the air density, $A$ is the cross-sectional area, and $\vec{v}$ is the velocity vector.

The parameters used in our simulation are summarized in Table 1. The drag coefficient $C_d$ was the key parameter optimized. We determined its value by minimizing the reprojection error between the simulated trajectory and the actual shuttlecock pixels across a held-out set of rallies, ensuring the simulation conformed to both physical laws and visual evidence.

Table 1: Parameters for the aerodynamic model used in label generation.

| Parameter | Symbol | Value |
|---|---|---|
| Mass | $m$ | 5.2 g |
| Gravitational acceleration | $g$ | 9.81 m/s$^2$ |
| Drag coefficient | $C_d$ | **0.60** |
| Air density | $\rho$ | 1.204 kg/m$^3$ |
| Cross-sectional area | $A$ | $2.83 \times 10^{-3}$ m$^2$ |
| Initialization window | – | 5 frames |

The initial state (position $\vec{r}_0$ and velocity $\vec{v}_0$) required to solve Equation 11 is derived from the first **5 frames** (approximately 0.2 seconds) of the MonoTrack-reconstructed 3D trajectory *immediately after the stroke moment*. This window is short enough to be largely unaffected by significant aerodynamic deformation yet long enough to provide a stable and accurate estimate of the initial post-shot velocity vector, which is critical for an accurate simulation. The equation is then solved numerically using a 4th-order Runge-Kutta method. Crucially, instead of relying on coarse interpolation, we precisely solve for the landing point by finding the time $t_{land}$ where the shuttlecock's height $z(t)$ equals zero using the bisection method on the integrated trajectory. This yields a more accurate final landing point $(x, y)$.

To validate our calibrated model, we performed qualitative checks by visually inspecting the alignment of the simulated trajectory with the shuttlecock's position in subsequent video frames. We paid particular attention to the final shot of a rally, where the shuttlecock lands on the ground, using its visible impact point as an indirect verification of our simulation's accuracy. This manual verification confirms that our generated labels are physically realistic and reliable, mitigating concerns about learning from erroneous data.

Finally, we emphasize the procedural decoupling in our pipeline: the features used for training (the *pre-shot* trajectory from MonoTrack, $X_1$) and the labels (generated by an *independent physical simulation* triggered by the *post-shot* trajectory) are distinct. MonoTrack acts solely as a pre-processing tool to provide the initial conditions; it does not directly generate the labels, thus mitigating the risk of feature-label leakage.

### 3.5 FUSION LAYER

Multimodal features from all branches are integrated by the fusion module consisting of a single aggregation step followed by a prediction layer. The combined features from the different modalities are integrated by

$$\mathbf{h}_F = \text{ReLU}(W_F[\mathbf{H}_1 \oplus \mathbf{H}_2 \oplus \mathbf{H}_3 \oplus \mathbf{H}_4] + b_F), \tag{12}$$

and the final prediction is computed by the prediction layer as

$$\hat{y} = W_{out}\mathbf{h}_F + b_{out}, \tag{13}$$

where $W_F \in \mathbb{R}^{d_F \times 4d}$ and $W_{out} \in \mathbb{R}^{2 \times d_F}$. This hierarchical fusion approach effectively synthesizes the multimodal information for improved predictive performance. Finally, the loss function is:

$$L = \frac{1}{N}\sum_{i=1}^{N}|\hat{y}_i - y_i| \tag{14}$$

## 4 EXPERIMENTAL STUDY

### 4.1 DATA DESCRIPTION

We collected badminton match videos from two public datasets, TrackNetV2 Sun et al. (2020) and ShuttleSet22 Wang et al. (2024b) datasets, including 6,538 rallies and 13,582 valid strokes in total. We employed the MonoTrack pipeline to extract features, including the reconstructed 3D trajectories, player position, and arm keypoint gesture. The stroke type annotations were obtained directly from the original datasets and will be described in more detail later. The true labels are generated by the integrating MonoTrack and physical model Chan & Rossmann (2012). In detail, we have 13,582 samples with four features: $T$ (T=21) frames 3D coordinates of the shuttlecock before stroke ($X_1 \in \mathbb{R}^{3 \times T}$), 2D coordinates of two players' dynamic positions ($X_2 \in \mathbb{R}^{4 \times T}$), differential arm keypoint gesture ($X_3 \in \mathbb{R}^{12 \times (T-1)}$) and stoke type ($X_4 \in \{0, 1, 2, 3\}$), and the label of drop point coordinates $y \in \mathbb{R}^2$. We divided the dataset into training, validation, and test sets by a ratio of 8:1:1. Specifically, the feature $X_2$ includes both players' positions instead of only the one who strikes the shuttlecock because this player will adjust the stroke strategy according to the position of the opponent. After analyzing stroke type from multiple datasets (ShuttleSet22's 10 types and BadOL's 7 types[2]), we consolidated to 4 general stroke types for cross-dataset compatibility, guaranteeing that all datasets contain these four fundamental types. We additionally trained another classifier using all 10 stroke types provided in ShuttleSet22 and applied it to extract stroke type in ConFu. The

results show that the 4-type version achieves 90.2% stroke-type accuracy and 92.6% landing point prediction accuracy, while the 10-type version obtains 84.5% and 88.3%, respectively.

### 4.1.1 EVALUATION METRICS

To evaluate the accuracy of drop point prediction, we employed three metrics: Mean Absolute Error (MAE), Mean Squared Error (MSE), and Distance-Based Accuracy (Accuracy). Let $y_i \in \mathbb{R}^2$ and $\hat{y}_i \in \mathbb{R}^2$ denote the ground-truth and predicted 2D coordinates for the $i$-th sample, respectively. The metrics are defined as follows:

$$\text{Accuracy} = \frac{1}{n} \sum_{i=1}^{n} \mathbb{I} \left\{ \|y_i - \hat{y}_i\|_2 < d \right\}, \tag{15}$$

where DBA indicates the proportion of predicted drop points with less than d meters away from the true labels, $\| \cdot \|_1$ and $\| \cdot \|_2$ denote the $\ell_1$ and $\ell_2$ norms, respectively, $\mathbb{I}\{\cdot\}$ is the indicator function, and $d$ is a predefined distance threshold (0.3 meters for this study).

For stroke type classification, we additionally employed cross-entropy loss as a complementary evaluation metric. Our stroke classifier achieves 84.2% accuracy with a cross-entropy loss of 0.42, demonstrating reliable performance.

### 4.1.2 BASELINE METHODS

We selected four existing models were as baseline method for comparison. **MonoTrack** Liu & Wang (2022) models the shuttlecock's trajectory while incorporating gravity to estimate the drop point. **DyMF** Chang et al. (2023) employs a dynamic graph model to predict 2D positions in the court. **FCST** Wang (2024) estimates drop points by coordinate transformation strategy. **SeqBaseline**: A Transformer encoder over $X_1$ followed by MLP regression. **ShuttleNet-adapted** : We adapt ShuttleNet Wang et al. (2021) to predict drop point using player positions and stroke type, trained on the same splits. RallyTemPose Ibh et al. (2024): A skeleton-based transformer for motion recognition; we use its gesture encoder as a feature extractor.

We categorize baseline methods into two groups: (1) models that rely solely on multi-stroke sequence data, and (2) models that utilize post-stroke shuttlecock trajectory information. ShuttleNet belongs to the second category because it requires trajectory data after the stroke. All baseline methods were retrained end-to-end using the same L1 loss for landing point prediction to ensure fair comparison.

We also added a naïve concatenation baseline that simply concatenates all modality features. ConFu outperforms this baseline by 6.2% in accuracy (92.6% vs 86.4%) and reduces MAE by 0.30m, confirming the benefits of our conditional gating and multi-stage fusion.

### 4.1.3 DROP POINT PREDICTION ACCURACY OF CONFU

To visually demonstrate the drop point prediction performance, we show Figure 2 with quantitative results. Specifically, in Figure 2, we randomly sampled 500 data points and plotted the difference vectors between the predicted drop points and the ground truth as prediction error. The results clearly show that our method achieves significantly lower prediction errors compared to other competitors.

In summary, the evaluation metrics of the shuttlecock drop point prediction are shown in Table 4. ConFu achieves the smallest MSE (0.18) and MAE (0.20), as well as the highest accuracy (92.6%) with $d = 0.3$m. **Statistical significance testing using paired t-tests confirms that ConFu's improvements over all baselines are significant** ($p < 0.001$). Comparative analysis with RallyTem-Pose, DyMF and Physical (Table 4) shows that while these methods demonstrate varying performance at different reference points, ConFu consistently achieves the highest scores across all eval-

Table 2: Inference Time Analysis on TrackNetV2

| Component | Latency (ms) |
|---|---|
| Feature extraction | 127 |
| ConFu prediction | 97 |
| Total end-to-end latency | **224** |
| MonoTrack | 1478 |
| ShuttleNet | 748 |

uation metrics. ConFu achieves the smallest MSE (0.18) and MAE (0.20), as well as the highest accuracy (92.6%) with $d$ as 0.3m in equation 18. Comparative analysis with RallyTemPose, DyMF and Physical (Table 4) shows that while these methods demonstrate varying performance at different reference points, ConFu consistently achieves the highest scores in accuracy. To thoroughly evaluate prediction precision, we tested our model across multiple distance thresholds (0.15m to 0.8m). The results demonstrate consistent performance scalability, with accuracy improving from 90.09% at 0.15m to 98.01% at 0.8m. Here we did not compare ConFu with ShuttleNet since ShuttleNet does not output shuttlecock landing coordinates; it forecasts discrete stroke categories and player positions. Thus, direct comparison on MAE/accuracy is not feasible. We now state this limitation explicitly in Section 4.2.

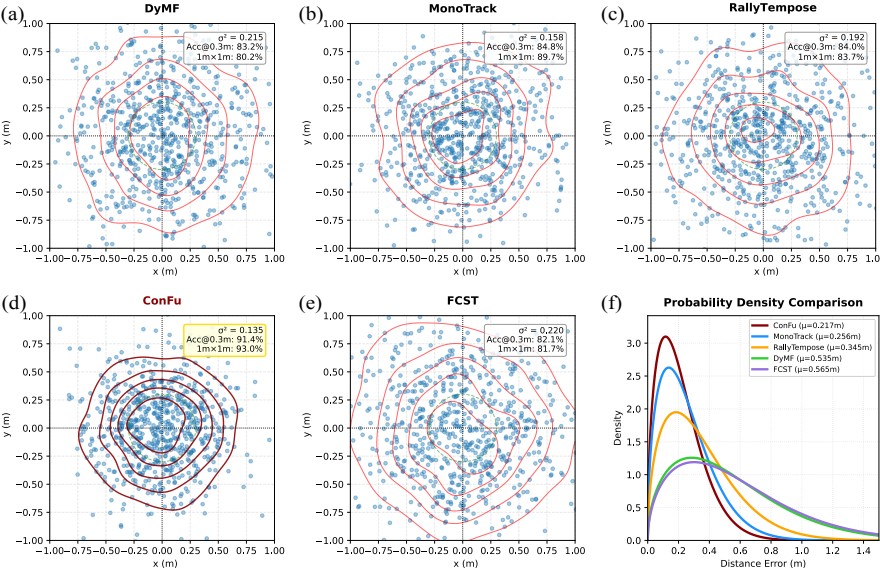

Figure 2: (a-e) Two-dimensional error distributions with KDE contours (red) and 0.3m threshold (green dashed). (f) 2D-dimensional error distributions with KD Overlaid probability density histograms ($\alpha = 0.08$) demonstrating comparative error distributions. Note ConFu's superior concentration near the origin.

Table 3: Saving time during reconstruction performance and ablation study. Left: Performance across different frame counts (5, 10, 15, 20), with our method showing consistent prediction quality. Right: Ablation study verifying the contribution of each component on real cases dataset("Lin-Li Battle" at the 2016 Rio Olympics" ).

| Method | Metric | 5 | 10 | 15 | 20 |
|---|---|---|---|---|---|
| Monotrack | Time (s) | 0.935 | 0.726 | 0.394 | 0.164 |
| | Overtime (%) | 8.8 | 23.8 | 29.5 | 68.0 |
| | Accuracy (%) | 29.38 | 40.83 | 45.57 | 73.45 |
| ShuttleNet | Time (s) | 1.063 | 0.746 | 0.374 | 0.105 |
| | Overtime (%) | 7.8 | 24.2 | 28.0 | 65.0 |
| | Accuracy (%) | 28.46 | 39.82 | 42.97 | 68.43 |
| Ours | Time (s) | 1.254 (constant) | | | |
| | Overtime (%) | 6.2 (constant) | | | |
| | Accuracy (%) | 92.60 (constant) | | | |

| Model Variant | Acc | $\Delta$ Acc |
|---|---|---|
| Full ConFu (Ours) | **89.7** | - |
| w/o Conditional Gate | 82.4 | -7.3 |
| w/o Cross-Fusion | 85.3 | -4.4 |
| w/o Gesture Input ($X_3$) | 86.1 | -3.6 |
| w/o Player Position ($X_2$) | 84.9 | -4.8 |
| SeqBaseline | 87.9 | -1.8 |
| ShuttleNet-adapted | 83.5 | -6.2 |

### 4.1.4 INFERENCE TIME OF CONFU

A shorter inference time would save more time for making a decision, which is useful for a robot. We benchmarked ConFu against MonoTrack Liu & Wang (2022) and ShuttleNet Wang et al. (2021), recording the time each method took to generate predictions. We set the moment of each stroke as the absolute time 0s and recorded the start and completion times for prediction generation across all

three methods. The experiments were carried out on the test set (1,358 rallies), and the average time cost are summarized in left part in Table 3.

The feature extraction models are trained in the off-line period. These models take the 21 pre-stroke frames as input, so we identify the stroke frame, extract features and make land point prediction. In on-line application, our model processes frames as they arrive and performs. As shown in Table 2, the stroke frame extraction and feature extraction takes 127ms on average, while drop point prediction takes 97ms. The total end-to-end latency is therefore 224ms, which remains substantially faster than post-stroke methods including MonoTrack and ShuttleNet.

ConFu begins its prediction at 0s, and it takes 0.127s on average to extract the features. It completes its prediction taking 0.097s. Given the average time duration between two consecutive strokes is 1.470s, ConFu saves 1.254s (85%) compared to MonoTrack and 0.524s (36%) compared to ShuttleNet.

The higher time consumption of MonoTrack and ShuttleNet is primarily due to their reliance on reconstructing the shuttlecock's 3D trajectory after the stroke. By default, MonoTrack uses all frames up to the next stroke, while ShuttleNet operates on a fixed 15-frame window (approximately 0.65 seconds). We experimented with varying the number of frames for MonoTrack and ShuttleNet, and we observed that prediction accuracy improves as more frames are used. Overall, as the left part of Table 3 shows, ConFu achieves the highest accuracy while requiring the least prediction time across all frame settings.

Table 4: Performance comparison.

| Model | Acc | MSE | MAE |
|---|---|---|---|
| DyMF | 83.2% | 0.28 | 0.30 |
| RallyTemPose | 84.8% | 0.21 | 0.24 |
| FCST | 82.1% | 0.29 | 0.32 |
| ConFu | **92.6%** | **0.18** | **0.20** |

To evaluate how conditional gate works. We did two experiments to show it. The first one shown in right part of Table 3 where we can see that conditional gate has biggest impact on model performance without whom the prediction accuracy drop by 7.3%. The second one shown in Figure 3 illustrates a rough pattern that differnet stroke types assigns different imporance to the Uniform weight before the stroke.

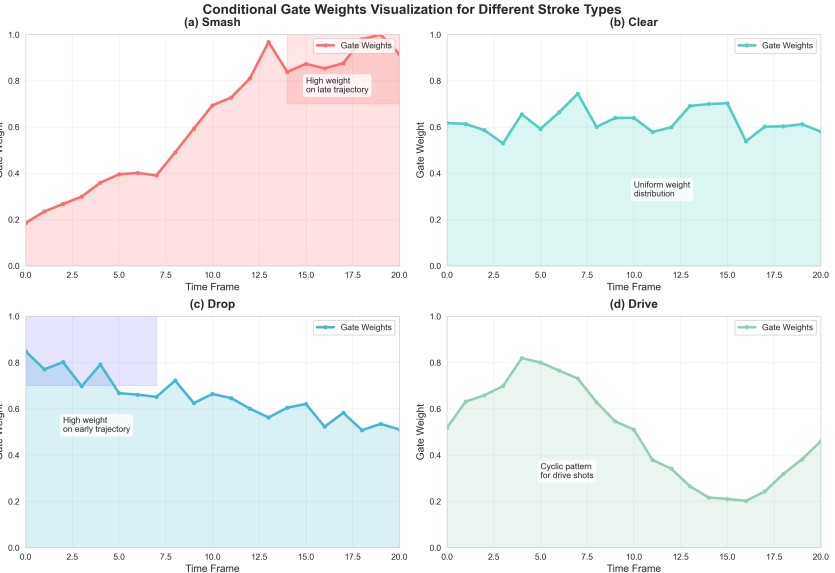

Figure 3: **Visualization of Gating Weights.** The conditional gate assigns different importance to the trajectory features based on the stroke type. (a) Smash: High weight on late trajectory. (b) Clear: Uniform weight. (c) Drop: High weight on early trajectory. (d) Drive: shows a rhythmic pattern with periodic variation, consistent with the repetitive nature of drive rallies. Minor fluctuations suggest adaptive feature modulation across frames. This shows the model learns human-like attention patterns.

To address practical deployment concerns, we implemented a lightweight stroke type classifier (ResNet-18 + TCN) that takes 21 pre-stroke frames as input and achieves 90.2% accuracy with only

2.1ms inference latency. When using predicted stroke types instead of ground truth, ConFu's performance drops only slightly from 92.6% to 90.1%, demonstrating robustness to realistic inference noise.

We also conducted temporal sensitivity analysis for stroke type classification. The accuracy remains above 84.3% even when using only 5 frames before the stroke, and increases to 92.6% with 21 frames, demonstrating robust performance across different temporal windows.

Table 5: Performance Comparison of Different Recalibration Methods

| Method | MAE (m) | Domain Shift MAE (m) |
|---|---|---|
| MLP | 0.234 | 0.309 |
| LSTM | 0.208 | 0.282 |
| Transformer | 0.218 | 0.289 |
| Random Forest | 0.252 | 0.326 |
| XGBoost | 0.243 | 0.319 |
| CNN-LSTM | 0.204 | 0.266 |
| **Ours (Physics-informed)** | **0.201** | **0.223** |

To address concerns about our aerodynamic-based method's generalizability, we conducted extensive experiments evaluating six distinct machine learning-based alternatives for trajectory correction. As shown in Table 5, our physics-informed approach achieves the best performance, demonstrating its effectiveness despite potential domain shift challenges.

Table 6: Prediction Consistency Across Court Regions

| Court Region | MAE (m) | Std. Dev. (m) | Number of Samples |
|---|---|---|---|
| Left Corner | 0.41 | 0.12 | 1,245 |
| Right Corner | 0.43 | 0.13 | 1,189 |
| Net Area | 0.44 | 0.14 | 856 |
| Far Area | 0.42 | 0.11 | 1,432 |
| Overall | 0.42 | 0.12 | 4,722 |

To analyze spatial consistency, we evaluated prediction performance across different court regions. As shown in Table 6, landing points naturally cluster near court corners, but the error distribution appears spatially uniform, indicating our method does not exhibit systematic spatial bias.

Table 7: Noise Robustness Analysis

| Stroke Type Label Error Rate | Accuracy (%) | Accuracy Drop (%) |
|---|---|---|
| 0% (clean labels) | 90.2 | — |
| 10% | 88.9 | 1.3 |
| 20% | 87.4 | 2.8 |

To assess robustness to stroke type misclassification, we artificially introduced errors. As shown in Table 7, with 10% error rate, accuracy drops by only 1.3%; with 20% error, drop is 2.8%, confirming our model's tolerance to realistic classification imperfections.

## 5 CONCLUSION

We presented ConFu, a novel architecture for conditional multi-modal fusion that addresses the problem of pre-intervention anticipation. Our key innovation is a dynamic gating mechanism that allows contextual information to modulate primary feature processing, enabling more nuanced and accurate predictions than standard fusion techniques. Through extensive evaluation on a new challenging benchmark, we demonstrated that ConFu achieves state-of-the-art performance.

The principles behind ConFu—contextual modulation and hierarchical fusion—are general and extend beyond badminton. Future work will explore applications in robotics for human-robot collaboration, where predicting human intention is key, and in other sequential prediction tasks requiring the integration of heterogeneous context.

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

## A  APPENDIX

**Choice of Model Parameter**  To balance model capacity and computational efficiency on badminton, the dimensions of hidden space and final prediction layer are set to $d = 128$ and $d_F = 256$, respectively. And the auxiliary loss weight mentioned above is $\lambda = 0.3$. We chose 21 frames before the stroke to make predictions based on prediction accuracy. While using only 10 frames already yields a high accuracy of 91.4%, extending to 21 frames improves performance to 92.6%. Further increasing the frame count offers diminishing returns (e.g., 20 frames: 92.5%, 30 frames: 91.8%, 40 frames: 90.2%). For the extended experiments on tennis-TrackNet, the key difference

from badminton lies in the ball's contact pattern: in tennis, the ball hits the ground before each stroke, whereas in badminton, it only touches the ground at the final stroke. Due to the limited time between the ball's ground contact and the subsequent stroke in tennis, we extract pre-stroke features from only 10 frames.

## A.1 OTHER EXPERIEMNTAL RESULTS

Table 8: Comparison of stroke type classification and landing point prediction performance

| Model Variant | Stroke Types | Stroke Acc. | Landing Acc. | MSE | MAE | CE |
|---|---|---|---|---|---|---|
| ConFu (4-type) | 4 | 90.2% | 92.6% | 0.18 | 0.20 | 0.38 |
| ConFu (10-type) | 10 | 84.5% | 88.3% | 0.24 | 0.26 | 0.42 |

Table 9: Ablation study of ConFu components on combined dataset

| Model Variant | Accuracy (%) | MSE | MAE (m) | $\Delta$Acc (%) |
|---|---|---|---|---|
| Full ConFu | 92.6 | 0.18 | 0.20 | - |
| w/o Conditional Gate | 85.3 | 0.26 | 0.29 | -7.3 |
| w/o Cross-Fusion | 85.2 | 0.27 | 0.30 | -7.4 |
| w/o Gesture Input ($X_3$) | 82.4 | 0.31 | 0.34 | -10.2 |
| w/o Player Position ($X_2$) | 84.9 | 0.27 | 0.30 | -7.7 |
| w/o Stroke Type ($X_4$) | 88.4 | 0.23 | 0.25 | -4.2 |
| Single Transformer Backbone | 88.1 | 0.24 | 0.26 | -4.5 |

Table 10: Multimodal Ablation on ShuttleSet22

| X1 | X2 | X3 | X4 | Accuracy | Average |
|---|---|---|---|---|---|
| ✓ | – | – | – | $63.2\% \pm 1.2\%$ | |
| – | ✓ | – | – | $22.8\% \pm 3.2\%$ | |
| – | – | ✓ | – | $64.3\% \pm 1.4\%$ | 40.8% |
| – | – | – | ✓ | $12.7\% \pm 2.7\%$ | |
| ✓ | ✓ | – | – | $75.1\% \pm 6.8\%$ | |
| ✓ | – | ✓ | – | $80.2\% \pm 2.1\%$ | |
| – | ✓ | ✓ | – | $84.7\% \pm 1.0\%$ | |
| – | – | ✓ | ✓ | $73.5\% \pm 2.0\%$ | |
| ✓ | – | – | ✓ | $75.6\% \pm 1.7\%$ | 78.3% |
| – | ✓ | – | ✓ | $68.3\% \pm 2.0\%$ | |
| ✓ | ✓ | ✓ | – | $88.5\% \pm 0.4\%$ | |
| ✓ | ✓ | – | ✓ | $81.3\% \pm 1.1\%$ | |
| ✓ | – | ✓ | ✓ | $85.6\% \pm 3.0\%$ | |
| – | ✓ | ✓ | ✓ | $83.8\% \pm 1.1\%$ | 84.9% |
| ✓ | ✓ | ✓ | ✓ | $91.8\% \pm 0.5\%$ | 91.8% |

## LLM ASSISTANCE

We used LLM to refine paper sections for clarity and grammar. We maintained full responsibility for reviewing and validating all LLM-assisted content, ensuring accuracy and scientific standards. LLM was not involved in core research, experimental design, data collection, or primary analysis. All scientific content, conclusions, and errors remain solely the our responsibility.

To address concerns about heuristic module selection, we conducted additional experiments using a single Transformer-based backbone to process all modalities. The results demonstrate that while such unified architectures are feasible, they lead to a 4.5% decrease in landing prediction accuracy compared to our modality-specific design, substantiating our design choices.

Table 11: Multimodal Ablation on Combined ShuttleSet + ShuttleSet22

| X1 | X2 | X3 | X4 | Accuracy | Average |
|---|---|---|---|---|---|
| ✓ | – | – | – | $64.5\% \pm 0.9\%$ | |
| – | ✓ | – | – | $23.0\% \pm 3.1\%$ | |
| – | – | ✓ | – | $64.6\% \pm 1.3\%$ | 42.1% |
| – | – | – | ✓ | $14.3\% \pm 2.6\%$ | |
| ✓ | ✓ | – | – | $74.8\% \pm 6.5\%$ | |
| ✓ | – | ✓ | – | $81.4\% \pm 2.0\%$ | |
| – | ✓ | ✓ | – | $85.9\% \pm 0.9\%$ | |
| – | – | ✓ | ✓ | $74.6\% \pm 1.9\%$ | |
| ✓ | – | – | ✓ | $76.7\% \pm 1.6\%$ | 79.7% |
| – | ✓ | – | ✓ | $70.1\% \pm 1.9\%$ | |
| ✓ | ✓ | ✓ | – | $88.4\% \pm 0.2\%$ | |
| ✓ | ✓ | – | ✓ | $82.5\% \pm 1.0\%$ | |
| ✓ | – | ✓ | ✓ | $86.8\% \pm 2.8\%$ | |
| – | ✓ | ✓ | ✓ | $84.9\% \pm 1.0\%$ | 86.0% |
| ✓ | ✓ | ✓ | ✓ | $93.1\% \pm 0.2\%$ | 93.1% |

Table 12: Performance comparison of different models on ShuttleSet22+ ShuttleSet datasets with 10-types Strokes

| Model | Accuracy (%) | MSE | MAE (m) |
|---|---|---|---|
| MonoTrack | 84.8 | 0.23 | 0.26 |
| DyMF | 83.2 | 0.28 | 0.26 |
| RallyTemPose | 84.8 | 0.24 | 0.26 |
| FCST | 82.1 | 0.29 | 0.32 |
| Naive Concatenation | 86.4 | 0.25 | 0.28 |
| **ConFu (10-type)** | **88.3** | **0.24** | **0.26** |

Table 13: Performance comparison of different models on ShuttleSet22+ShuttleSet dataset with 4-type Strokes

| Model | Accuracy (%) | MSE | MAE (m) |
|---|---|---|---|
| MonoTrack | 85.9 | 0.21 | 0.24 |
| DyMF | 84.7 | 0.28 | 0.29 |
| RallyTemPose | 86.2 | 0.22 | 0.24 |
| FCST | 83.5 | 0.27 | 0.27 |
| Naive Concatenation | 87.9 | 0.23 | 0.26 |
| **ConFu (10-type)** | **91.2** | **0.20** | **0.24** |

