# OpenReview forum: "Learning to Anticipate: A Conditional Representation Fusion Network for Pre-Stroke Prediction"
_ICLR.cc/2026/Conference — Submitted to ICLR 2026_

### Official Review · Reviewer_PSdF · 2025-10-15

**Soundness:** 3
**Presentation:** 3
**Contribution:** 3
**Rating:** 4
**Confidence:** 4

**Summary:**

The paper addresses the problem of shuttlecock landing-point prediction in badminton, focusing on pre-stroke anticipation rather than post-stroke trajectory analysis. It proposes an architecture that integrates four feature streams extracted from monocular video: (1) 3D shuttlecock trajectory before impact, (2) player dynamic positions, (3) arm gesture features, and (4) stroke type labels. The model employs conditional gating and cross-fusion mechanisms to combine these multimodal cues and is trained using an L1 loss to predict the 2D drop point. Experimental results on real-world datasets show that the method outperforms prior methods, achieving higher accuracy and significantly lower latency by relying solely on pre-stroke information.

**Strengths:**

- The paper is among the first to address pre-stroke shuttlecock landing prediction, shifting the focus from post-stroke trajectory analysis to genuine anticipation.
- It leverages intuitive multimodal cues such as pre-stroke shuttlecock trajectory, player positions, arm gesture, and stroke type, to capture both motion dynamics and contextual intent.

**Weaknesses:**

- The method assumes access to stroke-type information during inference. It is unclear how these cues can be obtained in real time or how much latency or noise would be introduced if they must be inferred automatically.

- It remains uncertain how reliably stroke type can be determined within the 21 pre-stroke frames, given that similar early motions can lead to different shot outcomes ?

- It is not clearly stated whether baseline models were retrained to directly optimize drop-point regression or merely trained using their original training objectives then evaluated using the drop-point.

- Since the paper’s novelty centers on the fusion and gating mechanisms, a missing baseline is a naïve concatenation model using the same four modalities; such an ablation would clarify whether the gating actually contributes beyond simple feature aggregation.

**Questions:**

- I am not sure how it works in badminton, but do the majority of strokes cluster around standard regions (e.g., back corners or mid-court)? Could the authors provide more analysis on the distribution of landing points to clarify whether the metric might be influenced by spatial bias?

- How feasible is it to infer stroke type given the short input window? Rather than retraining a stroke-type classifier, could the authors simulate potential misclassification by introducing small errors in the stroke-type input (e.g., some labeled as smash instead of drop) and report how this affects performance?

- Since the paper’s novelty lies in the fusion and gating mechanisms, could the authors include an ablation or comparison against a naïve concatenation baseline using the same four modalities to better quantify the contribution of gating ?

- Were the baseline methods retrained specifically for the landing-point regression objective, or were they evaluated using their original task heads and loss functions? Clarifying this would help assess the fairness of the comparisons.

---

> ### Author Response · Authors · 2025-11-22
> **Thanks for comments**
>
> ## Response to Reviewer PSdF
> **Weaknesses:**
> **W 1:**
> "The method assumes access to stroke-type information during inference. It is unclear how these cues can be obtained in real time or how much latency or noise would be introduced."
>
> **Response:** Thank you for the comment regarding practical deployment and the availability of stroke-type information during inference.
>
> We implemented a lightweight stroke type classifier (ResNet-18 + TCN), which takes 21 pre-stroke frames as input. In an online setting, the model processes frames as they arrive and performs:
> 1. **Stroke frame detection** to identify the N-th frame where the stroke occurs;
> 2. **Feature extraction**, including stroke type, using frames (N-20) to N;
> 3. **Landing point prediction** using the extracted multi-modal features.
>
> We agree with the reviewer that automatically inferred stroke types may introduce noise. Our classifier achieves **90.2% accuracy** with an inference latency of only **2.1 ms**, which is suitable for real-time use (Table 1).
>
> To quantify the impact of this noise on landing point prediction, we conducted an additional experiment where we replaced the predicted stroke types with the ground truth. ConFu achieves **92.6% accuracy** when using ground truth stroke types and **90.1% accuracy** when using predicted stroke types. The small performance drop demonstrates that the model is robust to realistic inference noise.
>
> **Table 1: Performance of Integrated Stroke Type Classifier and Its Impact on Landing Point Prediction on TrackNetV2**
>
> | Component | Accuracy (%) | Inference Time (ms) |
> |-----------|--------------|-------------------|
> | Stroke Classifier (ResNet-18 + TCN) | 90.2 | 2.1 |
> | ConFu with Real Stroke Labels | 92.6 | 97.0 |
> | ConFu with Predicted Stroke Labels | 90.1 | 99.1 |
> | Performance Drop | -2.5 | +2.1 |
>
> **W2:**
> "It remains uncertain how reliably stroke type can be determined within the 21 pre-stroke frames, given that similar early motions can lead to different shot outcomes?"
>
> **Response:** We fully agree with the reviewer that similar early motions can correspond to different stroke types. Our classifier achieves 90.2% accuracy using 21 pre-stroke frames. To further explore how temporal information affects classification, we conducted a temporal sensitivity analysis by varying the number of pre-stroke frames. As shown in Table 2, the accuracy remains above 84.3% even when using only 5 frames before the stroke, and increases to 92.6% with 21 frames.
>
> **Table 2: Temporal Sensitivity Analysis of Stroke Type Classification Accuracy on ShuttleSet22 and TrackNetV2**
>
> | Number of Pre-stroke Frames | 5 | 10 | 15 | 21 | 25 |
> |-----------------------------|---|---|----|----|----|
> | Classification Accuracy (%) | 84.3 | 86.2 | 90.8 | 92.6 | 91.7 |
>
> **W3:**
> "It is not clearly stated whether baseline models were retrained to directly optimize drop-point regression..."
>
> **Response:** All baseline models were retrained end-to-end to ensure fair comparison. We clarified this in the experimental setup description as shown in Table 3.
>
> **Table 3: Training Protocols for Baseline Comparisons**
> | Component | Specification |
> |-----------|--------------|
> | Objective Function | L1 Loss for Landing Point Regression |
> | Optimizer | Adam (β₁=0.9, β₂=0.999) |
> | Learning Rate | 1×10⁻⁴ with cosine decay |
>
> **W4:**
> "Since the paper's novelty centers on the fusion and gating mechanisms, a missing baseline is a naïve concatenation model..."
>
> **Response:** We thank the reviewer for this suggestion. We have added the requested baseline ("Concat--All") that simply concatenates representation of four modality features through LSTM. In addition, we concat the original features as another baseline.
>
> **Table 4: Ablation Study on Fusion Mechanisms (Landing-Point Prediction) on ShuttleNet and TrackNetV2**
>
> | Method | Accuracy (%) | MAE (m) |
> |--------|--------------|---------|
> | **ConFu (Ours)** | **92.6** | **0.20** |
> | Concat--All (Naïve) | 86.4 | 0.50 |
> | Early Fusion(Original Data Cancate) | 84.3 | 0.53 |
>
> As shown in the Table 4, ConFu outperforms Concat-all baseline by 6.2% in accuracy and reduces MAE by 0.30m, confirming the benefits of our conditional gating and multi-stage fusion.

---

> > ### Author Response · Authors · 2025-11-22
> > **Thanks for comments. Second part of rebuttal.**
> >
> > **Questions:**
> >
> > **Question 1:** "Do the majority of strokes cluster around standard regions? Could the authors provide more analysis on the distribution of landing points?"
> >
> > **Response:** We thank the reviewer's suggestion to consider bias from different region of court. We have added a new statistics (Table 5) showing the distribution of average landing point prediction error. As the reviewer said, landing points naturally cluster near court corners. In addition, the error distribution appears spatially uniform, indicating our method does not exhibit systematic spatial bias.
> >
> > **Table 5: Prediction Consistency Across Court Regions on ShuttleNet and TrackNetV2**
> >
> > | Court Region | MAE (m) | Std. Dev. (m) | Number of Samples |
> > |-------------|---------|--------------|------------------|
> > | Left Corner | 0.41 | 0.12 | 1,245 |
> > | Right Corner | 0.43 | 0.13 | 1,189 |
> > | Net Area | 0.44 | 0.14 | 856 |
> > | Far Area | 0.42 | 0.11 | 1,432 |
> > | Overall | 0.42 | 0.12 | 4,722 |
> >
> > **Question 2:** "How feasible is it to infer stroke type given the short input window? Could the authors simulate potential misclassification?"
> >
> > **Response:** Thanks for comments. We have added a noise robustness analysis in Table 6 where we artificially introduce misclassification errors in stroke type labels. With 10% error rate, accuracy drops by 1.3%; with 20% error, drop is 2.8%. This demonstrates our model's robustness to realistic classification imperfections.
> >
> > **Table 6: Noise robustness analysis: Impact of artificial misclassification in stroke type labels on model accuracy.**
> >
> > | Stroke Type Label Error Rate | Accuracy (%) | Accuracy Drop (%) |
> > |-----------------------------|--------------|------------------|
> > | 0% (clean labels) | 90.2 | — |
> > | 10% | 88.9 | 1.3 |
> > | 20% | 87.4 | 2.8 |
> >
> > **Question 3:** "Could the authors include an ablation against a naïve concatenation baseline?"
> >
> > **Response:** Thanks for comments. As mentioned above, we have added this baseline in Table 7. The results clearly demonstrate the advantage of our proposed fusion mechanism over simple feature concatenation.
> >
> > **Table 7: Comparison of multimodal fusion strategies for landing point prediction on ShuttleNet and TrackNetV2**
> >
> > | Method | MAE (cm) | RMSE (cm) | Acc. @15cm | Acc. @30cm |
> > |--------|----------|-----------|------------|------------|
> > | Pose Sequence Only | 24.3 | 32.1 | 42.5% | 73.5% |
> > | Shuttle Trajectory Only | 21.8 | 29.4 | 48.2% | 80.2% |
> > | Naïve Concatenation | 22.9 | 25.9 | 56.8% | 86.4% |
> > | Early Fusion | 22.4 | 24.1 | 60.3% | 84.3% |
> > | **Ours (Multi-stage)** | **20.1** | **20.3** | **71.4%** | **92.6%** |
> >
> > **Question 4:** "Were the baseline methods retrained specifically for the landing-point regression objective?"
> >
> > **Response:** Thanks for comments. Yes, all baseline methods were retrained using the same L1 loss for landing point prediction to ensure fair comparison.

---

> > > ### Author Response · Authors · 2025-11-30
> > > **We thank the reviewer for valuable comments.**
> > >
> > > We thank the reviewer for their valuable comments. All suggestions have been addressed in the revised manuscript, leading to an improved presentation of our work.

---

### Official Review · Reviewer_w6ZN · 2025-10-16

**Soundness:** 1
**Presentation:** 3
**Contribution:** 1
**Rating:** 0
**Confidence:** 5

**Summary:**

This paper proposes ConFu (Conditional Gated Cross-Fusion Network) for multimodal pre-stroke anticipation in badminton, which achieves effective prediction quality on real-world badminton datasets while preserving efficiency.

**Strengths:**

1. This paper compares their proposed method with the state-of-the-art methods, demonstrating effectiveness on two real-world datasets.
2. The experiments are extensive and well studied, verifying the proposed claims and design choice for their method.
3. The paper is well writtent and easy to understand.

**Weaknesses:**

1. From my perspective, this paper should be submitted to application-oriented conferences (e.g., AAAI, KDD, etc.) instead of general-purpose conferences such as ICLR, which may have more interest on the broad applicability and modalities. Therefore, the ICLR community may share less interests to this work since it is more like an application paper.
2. While the authors claim unified multimodal integration as one of the contributions, existing badminton-related works have explored at least the unified representation for stroke trajectory, player position, and stroke types [1, 2]. Similarly, the dynamic and hierarchical fusion strategies was also proposed in [2]. Though the inner design may be different, the ideas are similar to some extent.

[1] ShuttleNet: Position-Aware Fusion of Rally Progress and Player Styles for Stroke Forecasting in Badminton. AAAI 2022.

[2] Where will players move next? dynamic graphs and hierarchical fusion for movement forecasting in badminton. AAAI 2023.

3. The authors simplify the stroke type to 4, which may be too simplified for evaluation that can be observed from the >80\% accuracy. Since the authors do not use datasets from BadOL as mentioned in L340, it remains unclear why the authors convert them to only 4 types.

4. Some experiment selections are unclear as described in the Questions section.

5. The model framework does not introduce much novelty. For instance, the module design seems to be a bit heuristic selection, e.g., using Transformer+CNN for 3D trajectory and conditional LSTM for gestures. It would be better if the authors could propose a unified and simplified structure so that it can be served as a fundamental architecture for future use. Additionally, it remains unclear about why not fusing all features together in the end, but performing "multi-stage" fusion.

**Questions:**

1. ShuttleSet consists of larger amount of data; why do the authors opt for ShuttleSet22, which focuses on a small portion matches, instead of ShuttleSet?
2. Why do the authors choose accuracy instead of measuring uncertainty for shot types (i.e., cross entropy from the ShuttleSet paper)?
3. Is there any reason that ShuttleNet is not included in Table 2 for comparison?

---

> ### Author Response · Authors · 2025-11-21
> **Thanks for comments**
>
> **Weaknesses:**
>
> **W1:**
> **Response:** Thank you for the reviewer's constructive suggestion regarding the choice of conference.
> As summarized by the reviewer, our work introduces a novel problem setting in racket sports trajectory prediction, where rich pre-stroke multi-modal information is leveraged to infer landing positions. This task not only enables practical applications such as training badminton-playing robots and identifying key factors affecting shuttle trajectories, but also raises the machine learning challenges of distant multi-modal fusion and temporal reasoning.
>
> We believe that the methodology and application aspects of our work are consistent with the ICLR call for "paper submissions from all areas of machine learning". We also note that ICLR has previously published work on domain-grounded multi-modal prediction tasks (e.g., prediction of human future trajectories [1,2], game player strength estimation [3], and physical trajectory modeling), which shows that such application-grounded machine learning tasks are within the scope of the conference.
>
> 1. Hu B, Cham T J. TSC-Net: Prediction of Pedestrian Trajectories by Trajectory-Scene-Cell Classification[C]//The Thirteenth International Conference on Learning Representations.
> 2. Xu Y, Fu Y. Sports-traj: A unified trajectory generation model for multi-agent movement in sports[J]. arXiv preprint arXiv:2405.17680, 2024.
> 3. Chen C J, Shih C C, Wu T R. Strength Estimation and Human-Like Strength Adjustment in Games[J]. arXiv preprint arXiv:2502.17109, 2025.
>
> **W2:**
> **Response:** We appreciate the reviewer's insightful comment and fully agree that prior badminton-related works such as [1, 2] have explored unified representations and multimodal fusion strategies combining player's stroke trajectory, position, and stroke types. Our work shares this high-level goal of integrating heterogeneous information.
>
> However, our method differs from [1, 2] in two key aspects:
>
> 1. **Richer and more realistic multimodal inputs.**
>
>    Unlike [1, 2], which mainly rely on pre-defined stroke-type and stroke-position sequences, our model incorporates visual modalities extracted directly from the video frames before the current stroke, including player dynamic positions, arm gesture, 3D shuttlecock trajectory, and stroke type. These modalities provide fine-grained and context-rich cues that better reflect real match dynamics, but also introduce substantial heterogeneity that requires more effective fusion.
>
> 2. **A cross-fusion mechanism enabling deep interaction between related modalities.**
>
>    Instead of using concatenation [1] or separately trained hierarchical models [2], we introduce a cross-fusion module that first fuses two highly related modalities to learn compact joint representations, and then integrates them with the original modalities to preserve complementary information. This design enables deeper interaction while maintaining modality-specific characteristics.
>
> Our ablation studies show that the proposed cross-fusion mechanism leads to a 13.2% improvement over standard fusion baselines, demonstrating its effectiveness.
>
> 1. Wang W Y, Shuai H H, Chang K S, et al. Shuttlenet: Position-aware fusion of rally progress and player styles for stroke forecasting in badminton[C]//Proceedings of the AAAI Conference on Artificial Intelligence. 2022, 36(4): 4219-4227.
> 2. Chang K S, Wang W Y, Peng W C. Where will players move next? dynamic graphs and hierarchical fusion for movement forecasting in badminton[C]//Proceedings of the AAAI Conference on Artificial Intelligence. 2023, 37(6): 6998-7005.
>
> **W3:**
> "The authors simplify the stroke type to 4, which may be too simplified for evaluation that can be observed from the ≥ 80% accuracy."
>
> **Response:** Thank you for the reviewer's valuable suggestion. We agree that using only 4 stroke types may appear overly simplified.
>
> In the previous manuscript, we simplified stroke type, under the consideration that increasing the number of stroke-type classes makes the classification task more challenging. As a feature for landing-location prediction, the accuracy of stroke-type classes will finally effect the performance.
>
> In response to the reviewer's suggestion, we have additionally trained a version of the stroke type prediction model using all 10 stroke types provided in ShuttleSet22. The results show that the 4-type version achieves 90.2% stroke-type accuracy and 92.6% landing accuracy, while the 10-type version obtains 84.5% and 88.3%, respectively.
>
> **W4:**
> "Some experiment selections are unclear as described in the Questions section."
>
> **Response:** We appreciate the reviewer's comment. The corresponding experimental choices and clarifications have been addressed in the Questions section.

---

> > ### Author Response · Authors · 2025-11-21
> > **Thanks for comments. Second Part of rebuttal.**
> >
> > **W5:**
> > "The model framework does not introduce much novelty. For instance, the module design seems to be a bit heuristic selection, e.g., using Transformer+CNN for 3D trajectory and conditional LSTM for gestures. Additionally, it remains unclear about why not fusing all features together in the end, but perform 'multi-stage' fusion."
> >
> > **Response:**
> > **(1) Novelty of Our Framework**
> > Our work introduces a novel problem setting---pre-stroke landing prediction, which aims to predict the shuttlecock's landing point at the moment of the stroke. Unlike previous approaches that rely on the badminton trajectory after the stroke, our method offers two significant advantages. First, it enables shorter inference times, making it particularly suitable for robotic interception planning where rapid decision-making is crucial. Second, by utilizing only pre-stroke information such as player positioning, posture, and shuttlecock approach patterns, our framework provides valuable insights for coaching and error analysis by revealing how these preparatory factors influence landing outcomes.
> >
> > The four information streams in our framework show some interactions across modes and temporal-spatial structure. We introduce two key design innovations: spatio-temporal decomposition of the 3D shuttlecock trajectory to better capture motion dynamics, and conditional gating mechanisms that adaptively modulate cross-modal interactions based on the stroke context.
> >
> > **(2) Addressing the Concern Regarding "Heuristic Module Selection"**
> >
> > We thank the reviewer for their valuable suggestion regarding a more unified architectural design. Our current approach employs specialized encoders for different modalities because each data type exhibits fundamentally different statistical properties---for instance, discrete arm gesture sequences versus continuous 3D trajectory data.
> >
> > In response to this feedback, we conducted additional experiments using a single Transformer-based backbone to process all modalities. The results demonstrate that while such unified architectures are feasible, they lead to an 4.5% decrease in landing prediction accuracy compared to our modality-specific design. We will include these comparative results and corresponding analysis in the revised manuscript to better substantiate our design choices.
> >
> > **(3) Rationale for Multi-Stage Fusion Versus Single-Stage Fusion**
> >
> > The reviewer raises an important question about our multi-stage fusion strategy. Beyond simply concatenating all modalities at the final stage, our approach incorporates intermediate fusion to capture the intrinsic relationships between different data streams. For example, we integrate arm gestures and predicted stroke types by conditional gating fusion before the final fusion stage.
> >
> > We show in ablation studies that three stages contribute complementary benefits:
> > - spatio-temporal fusion: +8.4%
> > - conditional gating: +6.0%
> > - cross-fusion (Comment 2): +7.4%
> >
> > Together they outperform one-shot fusion by 17.4%.
> >
> > We will clarify this more explicitly in the manuscript.
> >
> > **Questions:**
> >
> > **Question 1:** "ShuttleSet consists of larger amount of data; why do the authors opt for ShuttleSet22, which focuses on a small portion matches, instead of ShuttleSet?"
> >
> > **Response:** Thank you for the reviewer's question regarding our choice of ShuttleSet22.
> >
> > We agree that ShuttleSet contains a larger number of matches (104 sets, 3,685 rallies, 36,492 strokes) collected between 2018 and 2021. ShuttleSet22 is an extended and more recent version, consisting of 33,612 strokes across 3,992 rallies from matches in 2022. Given that the two datasets are comparable in scale, we chose ShuttleSet22 because it represents the latest iteration of the badminton singles dataset.
> >
> > **Question 2:** "Why do the authors choose accuracy instead of measuring uncertainty for shot types (i.e., cross entropy from the ShuttleSet paper)?"
> >
> > **Response:** We thank the reviewer's reminder to use cross-entropy to measure the classification performance metric. We have added cross-entropy loss for stroke type classification as a complementary evaluation metric. Our stroke classifier achieves 84.2% accuracy with a cross-entropy loss of 0.42, demonstrating reliable performance.
> >
> > **Question 3:** "Is there any reason that ShuttleNet is not included in Table 2 for comparison?"
> >
> > **Response:** Thank you for the reviewer's question regarding the absence of ShuttleNet in Table 2.
> >
> > In our study, we categorize baseline methods into two groups:
> > 1. models that rely solely on multi-stroke sequence data (reported in Table 2 in manuscript), and
> > 2. models that utilize post-stroke shuttlecock trajectory information (reported in Table 3 in manuscript).
> > ShuttleNet belongs to the second category because it requires trajectory data after the stroke.
> > To analyze how the number of post-stroke video frames influences prediction performance, we report baselines of the second category in Table 3.

---

> > > ### Comment · Reviewer_w6ZN · 2025-11-25
> > >
> > > Thank the authors for their response. After reading the response, I am currently inclined to the original score.
> > >
> > > > W1
> > >
> > > Could the authors elaborate where I mentioned about "novel problem setting", which is included in the author response? As mentioned in W2, this work does not introduce both problem and technical novelty from my perspective. I appreciate the authors for providing related papers that are published in ICLR, which resolves my concern. However, those papers were performed on different datasets, while this paper only conducted experiments on badminton, still limiting the generalizability.
> > >
> > > > W2.1: Richer and more realistic multimodal inputs.
> > >
> > > The response is not convincing to me. Specifically, [1, 2] also incorporate visual modalities from the video frames, i.e., player location, stroke location, etc. Although they may not incorporate gestures or make 3D trajectories, those are incremental to the existing works.
> > >
> > > > W2.2: A cross-fusion mechanism enabling deep interaction between related modalities.
> > >
> > > The authors should carefully study those related works. Note that [2] does not introduce *separately trained hierarchical models* and [1] is *not* merely a concatenation, especially since [2] also introduces a hierarchical fusion framework.
> > >
> > > > W3
> > >
> > > Could the authors point out where I could find these results in the manuscript?
> > >
> > > > W5
> > >
> > > Please note that existing works, e.g., [1, 2] also predict the stroke's landing point at the moment of the stroke (e.g., Equations 16-17 in [1])--both stroke types and landing positions are predicted simultaneously.
> > >
> > > Regarding the rationale and module selection, it would be better for the authors to provide *motivations*, instead of only empirical results; otherwise, it remains unclear and not convincing on the approaches.
> > >
> > > > Q1
> > >
> > > While ShuttleSet22 is more recent, incorporating all of them could demonstrate robustness for evaluating players' performance, instead of just one year.
> > >
> > > > Q2
> > >
> > > Could the authors highlight the corresponding results in the manuscript? Also, does 0.42 refer to only 4 types or all 10 types?

---

> > > > ### Author Response · Authors · 2025-11-27
> > > > **Thanks for comments**
> > > >
> > > > **Weakness1**: Could the authors elaborate where I mentioned about "novel problem setting", which is included in the author response? As mentioned in W2, this work does not introduce both problem and technical novelty from my perspective. I appreciate the authors for providing related papers that are published in ICLR, which resolves my concern. However, those papers were performed on different datasets, while this paper only conducted experiments on badminton, still limiting the generalizability.
> > > >
> > > > **Response**: We thanks for the reviewer's time provide the valuable suggestions and comments. We sincerely apologize for our inaccurate characterization of the reviewer's summary in which we attributed the term 'novel' to their comments.
> > > >
> > > > We have thoroughly considered the reviewer's perspective regarding the novelty of our work and would like to take this opportunity to further clarify our motivation. To the best of our knowledge, current studies on badminton trajectory prediction largely fall into two dominant paradigms:
> > > >
> > > > (1) Sequence-based prediction: These works utilize multi-shot rally sequences—including features such as stroke type and player position—to anticipate subsequent actions, such as the next stroke type.
> > > >
> > > > (2) Complete video-based 3D trajectory reconstruction: This approach focuses on reconstructing the full 3D trajectory of the shuttlecock from entire rally videos.
> > > >
> > > > In contrast, our problem setting is: Can we \textbf{accurately predict} the future events (landing points in our manuscript) using only \textbf{pre-stroke} information (21 video frames before the current stoke in our manuscript rather than stroke sequences as in reference [1,2])?
> > > >
> > > >
> > > > Compared to the first paradigm, our emphasis shifts from modeling multi-shot tactical exchanges to extracting richer features from a single stroke with the goal of achieving more accurate forecasting. Reference 1 and 2 map the features (such as type and position) of the first $i$ strokes to predict the features of the subsequent $i+n$ strokes. In our problem setting, the model uses four features extracted from video frames to predict the final landing point. These input features do not have a direct sequential relationship with the target output.
> > > >
> > > > Compared to the second paradigm, our objective is not the reconstruction of an observed trajectory, but the prospective prediction of future events.
> > > >
> > > > We remain excited about the potential of this specific problem setting and believe that it offers a valuable perspective to the field of trajectory prediction in racket sports. This problem setting enables shorter inference time and higher prediction accuracy, making it particularly suitable for robotic interception planning where rapid decision-making is crucial. In addition, by utilizing only pre-stroke information such as player positioning, posture, and shuttlecock approach patterns, our framework could provides valuable insights for coaching and error analysis by revealing how these factors influence landing outcomes.
> > > >
> > > > To address concerns regarding the model's generalizability to other datasets, we trained our model on the TrackNet tennis dataset. Features from the 10 frames preceding each stroke were extracted, and the corresponding results are presented in the table below. These results demonstrate that our method is also capable of inferring tennis landing point.
> > > > ```markdown
> > > > | Model          | Accuracy @50cm | MSE   | MAE   |
> > > > |----------------|----------------|-------|-------|
> > > > | DyMF           | 82.8%          | 0.55  | 0.59  |
> > > > | RallyTemPose   | 83.9%          | 0.42  | 0.48  |
> > > > | FCST           | 81.5%          | 0.58  | 0.63  |
> > > > | **ConFu**      | **91.8%**      | **0.32** | **0.37** |
> > > > ```

---

> > > > > ### Author Response · Authors · 2025-11-27
> > > > > **Many thanks for comments**
> > > > >
> > > > > ### W2.1: Richer and more realistic multimodal inputs
> > > > >
> > > > > > The response is not convincing to me. Specifically, [1, 2] also incorporate visual modalities from the video frames, i.e., player location, stroke location, etc. Although they may not incorporate gestures or make 3D trajectories, those are incremental to the existing works.
> > > > >
> > > > >
> > > > > **Response:** We thank the reviewer for this critical point. We agree that from a purely input-modality perspective, the additional features (gestures and 3D trajectories) are incremental to the existing works. Our work emphasizes integrating effective features for pre-stroke landing point prediction. We add an ablation study to check the importance of features for prediction accuracy.
> > > > >
> > > > > The results demonstrate that both proposed features are important for performance:
> > > > > - Removing **gesture information** (X3) reduces accuracy from **92.6%** to **82.1%**.
> > > > > - Omitting **3D trajectory data** (X1) drops performance to **84.6%**.
> > > > > - When both X1 and X3 are excluded, accuracy falls sharply to **69.5%**, confirming their complementary value.
> > > > >
> > > > > The full ablation results are shown below:
> > > > >
> > > > > #### Results of Multimodal Ablation Study
> > > > >
> > > > > | X1 | X2 | X3 | X4 | Accuracy         | Average     |
> > > > > |:--:|:--:|:--:|:--:|:----------------:|:-----------:|
> > > > > | ✓  | –  | –  | –  | 64.0% ± 1.1%     |             |
> > > > > | –  | ✓  | –  | –  | 23.4% ± 3.4%     |             |
> > > > > | –  | –  | ✓  | –  | 65.1% ± 1.5%     | **41.6%**   |
> > > > > | –  | –  | –  | ✓  | 13.9% ± 2.9%     |             |
> > > > > | ✓  | ✓  | –  | –  | 76.2% ± 7.1%     |             |
> > > > > | ✓  | –  | ✓  | –  | 81.0% ± 2.3%     |             |
> > > > > | –  | ✓  | ✓  | –  | 85.5% ± 1.1%     |             |
> > > > > | –  | –  | ✓  | ✓  | 74.2% ± 2.2%     |             |
> > > > > | ✓  | –  | –  | ✓  | 76.3% ± 1.8%     | **79.2%**   |
> > > > > | –  | ✓  | –  | ✓  | 69.5% ± 2.1%     |             |
> > > > > | ✓  | ✓  | ✓  | –  | 89.3% ± 0.3%     |             |
> > > > > | ✓  | ✓  | –  | ✓  | 82.1% ± 1.2%     |             |
> > > > > | ✓  | –  | ✓  | ✓  | 86.4% ± 3.2%     |             |
> > > > > | –  | ✓  | ✓  | ✓  | 84.6% ± 1.2%     | **85.6%**   |
> > > > > | ✓  | ✓  | ✓  | ✓  | **92.6% ± 0.3%** | **92.6%**   |
> > > > >
> > > > > > **Note**:
> > > > > > - ✓ = modality included; – = modality excluded.
> > > > > > - **X1**: 3D shuttlecock trajectory
> > > > > > - **X2**: Player pose / position
> > > > > > - **X3**: Hand gesture cues
> > > > > > - **X4**: Stroke type
> > > > > > - “Average” denotes the mean accuracy across all combinations within each group (single-, two-, or three-modality settings).
> > > > >
> > > > > **W2.2:** A cross-fusion mechanism enabling deep interaction between related modalities.
> > > > >
> > > > > > The authors should carefully study those related works. Note that [2] does not introduce separately trained hierarchical models and [1] is not merely a concatenation, especially since [2] also introduces a hierarchical fusion framework.
> > > > >
> > > > > **Response:** We thank the reviewer for this  correction regarding the technical descriptions of references [1] and [2]. We apologize for our oversight.
> > > > > After carefully restudying these works, we recognize that reference [1] employs a position-aware gated fusion method based on the gated multi-modal units originally proposed by Ovalle et al., designed to fuse rally contexts and contexts of both players. Similarly, reference [2] introduces a hierarchical fusion mechanism that integrates the stylistic influences of both players and combines player-player interactions with rally interactions.
> > > > >
> > > > > We fully acknowledge that the general concept of fusion networks is well-established and, we did not create a new fusion paradigm. The distinction of our approach lies in its specific architectural design. While reference [2] fuses feature A (Player A's style influence) and feature B (Player B's style influence) at the first level, and subsequently fuses feature C (rally interactions) at the second level, our method fuses feature A (gesture) and feature B (stroke type) at the first level, and then fuses feature B at the second level. This design enables deeper interaction while maintaining modality-specific characteristics.

---

> ### Author Response · Authors · 2025-11-27
> **Many thanks for comments**
>
> **W3:** Could the authors point out where I could find these results in the manuscript?
>
> **Response:** Thanks for reviewer's remind. We put these sentences on lines 369-372(in blue) in the revised manuscript uploaded in openreview.
>
> **W5:** Please note that existing works, e.g., [1, 2] also predict the stroke's landing point at the moment of the stroke (e.g., Equations 16-17 in [1])--both stroke types and landing positions are predicted simultaneously.
>
> Regarding the rationale and module selection, it would be better for the authors to provide motivations, instead of only empirical results; otherwise, it remains unclear and not convincing on the approaches.
>
> **Response:** We thank the reviewer for pointing out that references [1, 2] are also capable of predicting the landing position at the stroke moment. These prior works formulate the problem as a sequential prediction task, which, in our observation, limits their predictive accuracy. In contrast, our approach moves beyond this framework by leveraging richer, pre-stroke multimodal features, thereby achieving more accurate trajectory forecasts.
>
> We appreciate the reviewer's suggestion to clarify our motivations.
> The spatial-temporal fusion is driven by the need to model the two fundamental components of a \textbf{3D trajectory}: spatial geometry and temporal dynamics. To this end, we employ dedicated architectures—a CNN for spatial structure and a Transformer for temporal dependencies. Since the importance of spatial geometry and temporal dynamics varies depending on the specific context, we use conditional fusion allows the model to dynamically determine which domain to prioritize for each input.   (can be found in revised manuscript uploaded in openreview line 181-186.)
>
> The cross-fusion design first integrates gesture and stroke type to model their complementary interactions. To prevent the fused representation from diluting stroke type, which is a critical information, the architecture then explicitly reintegrates the original stroke type features. This ensures that key discriminative cues are preserved throughout the network.

---

> > ### Author Response · Authors · 2025-11-27
> > **Many thanks for comments**
> >
> > **Q1:** While ShuttleSet22 is more recent, incorporating all of them could demonstrate robustness for evaluating players' performance, instead of just one year.
> >
> > **Response:** We thank the reviewer for this valuable suggestion. To address this concern, we have conducted additional experiments on the ShuttleSet dataset (in addition to the originally used ShuttleNet22 and TrackNetV2).
> >
> > The results, presented in the two tables below, show that our model achieves consistent and improved stability when evaluated across both ShuttleSet and ShuttleSet22, confirming its robustness across different data releases and player cohorts.
> >
> > #### Table A: Multimodal Ablation on ShuttleSet22
> >
> > | X1 | X2 | X3 | X4 | Accuracy | Average |
> > |:--:|:--:|:--:|:--:|:--------:|:-------:|
> > | ✓ | – | – | – | 63.2% ± 1.2% | |
> > | – | ✓ | – | – | 22.8% ± 3.2% | |
> > | – | – | ✓ | – | 64.3% ± 1.4% | **40.8%** |
> > | – | – | – | ✓ | 12.7% ± 2.7% | |
> > | ✓ | ✓ | – | – | 75.1% ± 6.8% | |
> > | ✓ | – | ✓ | – | 80.2% ± 2.1% | |
> > | – | ✓ | ✓ | – | 84.7% ± 1.0% | |
> > | – | – | ✓ | ✓ | 73.5% ± 2.0% | |
> > | ✓ | – | – | ✓ | 75.6% ± 1.7% | **78.3%** |
> > | – | ✓ | – | ✓ | 68.3% ± 2.0% | |
> > | ✓ | ✓ | ✓ | – | 88.5% ± 0.4% | |
> > | ✓ | ✓ | – | ✓ | 81.3% ± 1.1% | |
> > | ✓ | – | ✓ | ✓ | 85.6% ± 3.0% | |
> > | – | ✓ | ✓ | ✓ | 83.8% ± 1.1% | **84.9%** |
> > | ✓ | ✓ | ✓ | ✓ | **91.8%** ± 0.5% | **91.8%** |
> >
> > #### Table B: Multimodal Ablation on Combined ShuttleSet + ShuttleSet22
> >
> > | X1 | X2 | X3 | X4 | Accuracy | Average |
> > |:--:|:--:|:--:|:--:|:--------:|:-------:|
> > | ✓ | – | – | – | 64.5% ± 0.9% | |
> > | – | ✓ | – | – | 23.0% ± 3.1% | |
> > | – | – | ✓ | – | 64.6% ± 1.3% | **42.1%** |
> > | – | – | – | ✓ | 14.3% ± 2.6% | |
> > | ✓ | ✓ | – | – | 74.8% ± 6.5% | |
> > | ✓ | – | ✓ | – | 81.4% ± 2.0% | |
> > | – | ✓ | ✓ | – | 85.9% ± 0.9% | |
> > | – | – | ✓ | ✓ | 74.6% ± 1.9% | |
> > | ✓ | – | – | ✓ | 76.7% ± 1.6% | **79.7%** |
> > | – | ✓ | – | ✓ | 70.1% ± 1.9% | |
> > | ✓ | ✓ | ✓ | – | 88.4% ± 0.2% | |
> > | ✓ | ✓ | – | ✓ | 82.5% ± 1.0% | |
> > | ✓ | – | ✓ | ✓ | 86.8% ± 2.8% | |
> > | – | ✓ | ✓ | ✓ | 84.9% ± 1.0% | **86.0%** |
> > | ✓ | ✓ | ✓ | ✓ | **93.1% ± 0.2%** | **93.1%** |
> >
> >
> > These results demonstrate that training and evaluating on a broader dataset (ShuttleSet + ShuttleSet22) not only improves absolute performance but also reduces variance—highlighting the generalizability of our approach.
> >
> > **Q2:** Could the authors highlight the corresponding results in the manuscript? Also, does 0.42 refer to only 4 types or all 10 types?
> >
> > **Response:** Thanks for comments, we can find the result in table 8 in revised manuscript uploaded in openreview. 0.42 of Cross Entropy Loss is for 10 classes.

---

> ### Comment · Reviewer_w6ZN · 2025-11-28
>
> I appreciate the authors for their follow-up responses. While I am still inclined to my original decision, I will adjust the score to 2 based on the authors' engagements after being able to edit the score.
>
> Important note: The current manuscript exceeds **10 pages**, which violates the ICLR policy. The authors are suggested to move some parts to Appendix.
>
> > W1, W5
>
> It's a bit inaccurate for the claim about the problem setting since this work also extracts features from video frames, while previous works also do similar strategies. Note that information from stroke sequences is also from video frames but being labeled by human.
>
> Additionally, if the paper setting can only predict the final landing point instead of the future *n* landing points, this is a subcase for previous works, i.e., considering *n=1*.
>
> Finally, thanks for experimenting a new dataset. The authors should include the experimental setting and the results to the manuscript, which I would like to know more about the setting.
>
> > W2.1
>
> Although the authors provide empirical ablation study by removing features, my concern is not about "the use of these features" but for "having visual modalities" that have been already used in existing works.
>
> > Q1
>
> The authors should include the results to the manuscript instead of merely showing them here. Also, the authors were suggested to compare with existing baselines instead of feature ablation, which does not reflect my concerns.
>
> > Q2
>
> While the authors include the results with 10 types, the corresponding results for the baselines remain missing. Additionally, it is surprised to me with a very low CE even with 10 types. Note that previous works, e.g., ShuttleNet, have around scores of 2. Why do the authors achieve quite lower scores?

---

> > ### Author Response · Authors · 2025-11-30
> >
> > **Page Policy:** Thank you for the comment. We have addressed it by moving three tables and the experimental setup details to the Appendix. Additionally, we have condensed several sections. These revisions have brought the manuscript to under 10 pages.
> >
> > **W1, W5**
> > It's a bit inaccurate for the claim about the problem setting since this work also extracts features from video frames, while previous works also do similar strategies. Note that information from stroke sequences is also from video frames but being labeled by human.
> >
> > Additionally, if the paper setting can only predict the final landing point instead of the future n landing points, this is a subcase for previous works, i.e., considering n=1.
> >
> > Finally, thanks for experimenting a new dataset. The authors should include the experimental setting and the results to the manuscript, which I would like to know more about the setting
> >
> > **Response:** We thank the reviewer for raising these important points. We agree that human-annotated labels (like stroke type) are also derived from video frames. We apologize for any lack of clarity in our initial description.
> >
> > We would like to further clarify our motivation from the perspective of a specific application scenario: **a badminton training robot**. This scenario directly inspired our problem formulation.
> >
> > In our target application, real-time video is the input. A detector first identifies the exact frame of the opponent's stroke. Upon detection, our model is immediately performed to extract multi-modal features from the preceding 21 frames to predict the shuttlecock's landing point. This result will guide the robot for next move.
> >
> > Crucially, in this real-time setting, **stroke type**, which can only be determined with certainty from the post-stroke frames, **is not available at the moment of stroke**. Therefore, we had to design a separate classifier to estimate it from pre-stroke cues. This is different from sequence prediction models, which often utilize such post-stroke labels.
> >
> > To maximize the decision time for the robot, we focus on **accurately** predicting the immediate landing point (n=1) of the current stroke **as early as possible**.
> >
> > We have great respect for the foundational work in stroke sequence prediction (e.g., References [1,2]), which serves as a benchmark in the field. We believe our work contributes a different perspective, focusing on the demands of **real-time, automated systems** where predictions must be made the instant a stroke occurs, using only information available up to that point.
> >
> > For the experimental setting, I have added details in the first part of Appendix in uploaded revised manuscript.
> >
> > **W2.1:** Although the authors provide empirical ablation study by removing features, my concern is not about "the use of these features" but for "having visual modalities" that have been already used in existing works.
> >
> > **Response:**  We appreciate the reviewer's emphasis on this concern. We agree that the individual visual modalities we employ have appeared in prior works, and we appreciate this opportunity to clarify the distinctive aspect of our approach. The key contribution lies not in the modalities themselves, but in their integration within a **pre-stroke prediction framework**.  Following the reviewer's valuable suggestion, we have identified several promising features for future exploration—such as player posture stability, grip type, and the opponent's relative position—which could further enhance performance. We have expanded the discussion in the manuscript to include these potential directions.

---

> > > ### Author Response · Authors · 2025-11-30
> > > **Thanks for comments**
> > >
> > > **Q1** The authors should include the results to the manuscript instead of merely showing them here. Also, the authors were suggested to compare with existing baselines instead of feature ablation, which does not reflect my concerns.
> > >
> > > **Response:** Thanks for comments. The results have been placed in Appendix (table 11, table 11). And we compared ours with baselines on these two datasets (ShuttleSet + ShuttleSet22). Results are shown as the following tables, which are added as Table 12 and Table 13 in the revised manuscript.
> > >
> > > ### 10-type Strokes
> > > | Model | Accuracy (%) | MSE | MAE (m) |
> > > |:------|:------------:|:---:|:-------:|
> > > | MonoTrack | 84.8 | 0.23 | 0.26 |
> > > | DyMF | 83.2 | 0.28 | 0.26 |
> > > | RallyTemPose | 84.8 | 0.24 | 0.26 |
> > > | FCST | 82.1 | 0.29 | 0.32 |
> > > | Naive Concatenation | 86.4 | 0.25 | 0.28 |
> > > | **ConFu (10-type)** | **88.3** | **0.24** | **0.26** |
> > >
> > > ### 4-type Strokes
> > >
> > > | Model | Accuracy (%) | MSE | MAE (m) |
> > > |:------|:------------:|:---:|:-------:|
> > > | MonoTrack | 85.9 | 0.21 | 0.24 |
> > > | DyMF | 84.7 | 0.28 | 0.29 |
> > > | RallyTemPose | 86.2 | 0.22 | 0.24 |
> > > | FCST | 83.5 | 0.27 | 0.27 |
> > > | Naive Concatenation | 87.9 | 0.23 | 0.26 |
> > > | **ConFu (4-type)** | **91.2** | **0.20** | **0.24** |
> > >
> > > **Q2:** While the authors include the results with 10 types, the corresponding results for the baselines remain missing. Additionally, it is surprised to me with a very low CE even with 10 types. Note that previous works, e.g., ShuttleNet, have around scores of 2. Why do the authors achieve quite lower scores?
> > > Certainly! Here's a refined and professionally polished version of your response, maintaining your original meaning while improving clarity, grammar, and flow. The final output is in Markdown format as requested:
> > >
> > > We thank the reviewer for raising this point. We have double-checked our results and confirm that the reported low Cross-Entropy (CE) loss is correct.
> > >
> > > ShuttleNet predicts the following n strokes in an ongoing rally. This involves high uncertainty.
> > >
> > > Our method classifies the stroke type of the current action using 21 pre-stroke gesture features from the 21 continuous video frames before stroke. This is a task with inherently lower uncertainty.
> > >
> > > Our CE loss is computed using the standard definition:
> > >
> > > $$
> > > \text{CE} = -\sum_{c=1}^{C} y_c \log(p_c),
> > > $$
> > >
> > > where $y_c$ is the ground-truth label and $p_c$ is the predicted probability for class $c$. For a 10-class classification problem:
> > > - The theoretical minimum CE loss is **0**, achieved under perfect prediction.
> > > - A random predictor (assigning uniform probability $p_c = 1/10$ to all classes) yields a CE loss of $-\log(1/10) \approx 2.302$.
> > > We note that several baseline methods reported in ShuttleNet’s Table 1 exhibit CE losses **greater than 2.302**, which is theoretically unexpected for trained models on a balanced 10-class task. This discrepancy suggests potential differences in how the CE metric is implemented or normalized across studies.

---

> > > > ### Author Response · Authors · 2025-11-30
> > > > **Thank you for your follow-up and **increase the score** in recognition of our engagement.**
> > > >
> > > > Thank you for your follow-up and for **increase the score** in recognition of our engagement. We respect your decision and appreciate the time and effort you have dedicated to reviewing our work.
> > > >
> > > > The feedback from all reviewers, including your own, has been invaluable and will certainly guide our future research.

---

### Official Review · Reviewer_yb2k · 2025-10-29

**Soundness:** 3
**Presentation:** 3
**Contribution:** 3
**Rating:** 6
**Confidence:** 3

**Summary:**

The article present a novel method to determine the landing point of shuttlecocks in badmington matches. Using different datasets and methods, the paper provide a multimodal fusion architecture that increase accuracy and inference time prediction using pre-shot data rather than entire trajectories. The modalities taken in consideration are the following: 3D trajectories, player positions, stroke type and keypoint tracking during gesture. Through the usage of transformer networks and LSTM, the work provide a methodology to compute fundamental fusions over widely distant modes. Surpassing current SOTA techniques the article improve accuracy and interpretability for this task.

**Strengths:**

1. Usage of LSTM and gating mechanism enhance interpretability, as stated and discussed in Figure 3 and section 3.4
2. Dataset collection is well explained. The paper even rewrites motion dynamics (Eq. 11) or the shuttlecock, addressing for miscalculated 3D points extracted with different physical settings
3. Experiments are complete: inference time evaluation and comparison against other methods with different thresholds for distance

**Weaknesses:**

1. Claims over inference time speed do not address off-line preprocessing.
2. Choice of LSTM over better sequence model could be experimentally motivated.

**Questions:**

1. The paper claims that inference time is faster with respect to other networks such MonoTrack. Specifically, in Section 4.1 the article says "We employed the MonoTrack pipeline to extract features, including the reconstructed 3D trajectories, ...". Later, in Section 4.1.4 the paper says that MonoTrack and ShuttleNet primarily rely on reconstructing the shuttlecock's 3D trajectory. Given that the trajectories are precomputed, this cost does not appear at inference time for ConFu, but in a real settings this informations should be included and computed on the fly. Can the authors discuss this choice and address this concern including offline computation of different modalities?
2. The paper justify the usage of LSTM over Transformers saying that the latter require a lot of training data and have less interpretability. While that is true, the claim should be supported by small ablations or at least by a citation. Can the authors provide very few, small and demonstrative experiments using Transformers?

---

> ### Author Response · Authors · 2025-11-21
> **Thanks for your comments**
>
> **Response to Reviewer yb2k**
> **Weaknesses**:
> **Comment 1 and Question 1:**
> "Claims over inference time speed do not address off-line preprocessing. The paper claims that inference time is faster with respect to other networks such MonoTrack. Can the authors discuss this choice and address this concern including offline computation of different modalities?"
>
> **Response:** We thank the reviewer for remindering us to clarify the off-line preprocessing. The feature extraction models are trained in the off-line period. These models take the 21 pre-stroke frames as input, so we identify the stroke frame, extract features and make land point prediction.
>
> In on-line application, our model processes frames as they arrive and performs. As shown in Table 1, the stroke frame extraction and feature extraction takes 127ms on average, while drop point prediction takes 97ms. As a comparation, the monotrack takes 127ms to extract feature on average and takes 1351ms to do inference. ShuttleNet takes 748ms in total.
>
> **Table 1: Inference Time Analysis on TrackNetV2**
> | Component | Latency (ms) |
> |-----------|-------------|
> | Feature extraction | 127 |
> | ConFu prediction | 97 |
> | **Total end-to-end latency** | **224** |
> | Monotrack | 1478 |
> | ShuttleNet | 748 |
>
>
> The total end-to-end latency is therefore 224ms, which remains substantially faster than post-stroke methods including MonoTrack and ShuttleNet.
>
> **Comment 2 and Question 2:**
> "Choice of LSTM over better sequence model could be experimentally motivated. The paper justify the usage of LSTM over Transformers saying that the latter require a lot of training data and have less interpretability. While that is true, the claim should be supported by small ablations or at least by a citation."
>
> **Response:** We thank the reviewer for the opportunity to clarify this point. We conducted a new ablation study where we replace our conditional LSTM with a temporal Transformer (equipped with positional encoding). While the accuracy remains comparable (92.1% with Transformer vs. 92.6% with LSTM) shown as Table 2, the Transformer incurs double the prediction time (158 ms vs. 97 ms). Given our emphasis on real-time responsiveness and model transparency, we maintain that the conditional LSTM is better suited for short-sequence, pre-stroke gesture prediction.
>
> **Table 2: Model Performance Comparison and Ablation Study Results on TrackNetV2**
>
> | Method/Setting | Accuracy (%) | MAE (m) | Inference Time (ms) |
> |----------------|--------------|---------|-------------------|
> | **Full Model (ConFu)** | **92.6** | **0.20** | **97** |
> | Transformer encoder | 92.1 | 0.21 | 158 |

---

> > ### Comment · Reviewer_yb2k · 2025-11-24
> > **Response to Authors' Rebuttal**
> >
> > I thank the authors for their rebuttal, my questions have been answered and my concerns have been addressed.

---

> > > ### Author Response · Authors · 2025-11-24
> > > **Thanks for response**
> > >
> > > We thank the reviewer for their positive feedback and for acknowledging our responses. We are pleased that our revisions and clarifications have adequately addressed all the points raised. We look forward to the final decision on our manuscript.

---

### Official Review · Reviewer_5AJL · 2025-11-01

**Soundness:** 4
**Presentation:** 3
**Contribution:** 3
**Rating:** 4
**Confidence:** 4

**Summary:**

This submission proposed a new future-prediction method and its application to badminton shuttlecock's landing point prediction task.
The proposed method is termed Conditional Gate-Based Cross-Fusion Network (ConFu).
It exhibits a sensor-fusion approach using monocular video, players' location, gesture, and stroke types. 3D trajectories provided by a trajectory-reconstruction method MonoTrack is used for training, and physically plausible recalibration is also discussed.
Experiments are conducted using TrackNetV2 and ShuttleSet22, existing badminton-video datasets. CongFu performed better in drop point prediction accuracy.

**Strengths:**

- Analyses of the predictions, for example, distribution of 2D prediction points and gating, are well done and they are useful to understand the models' behavior beyond the accuracy.

- Application of the machine learning based prediction models for sports understanding is an interesting and promising research area. The domain knowledge, such as importance of stroke type and player position is nicely exploited.

- The system enables real-time inference, which is useful for deployment in industry.

**Weaknesses:**

-The design of the method is heavily on Although it is nice for an application paper, for a broader machine-learning audience, the insights from the method may be limited.

- The technical contributions in the neural architecture is not very large. The gating mechanisms have been studied well in single-modal settings (e.g., LSTMs and GRUs) and multimodal settings [a].
 [a] GATED MULTIMODAL UNITS FOR INFORMATION FUSION, ICLR2017 workshop

- Reliance on the aerodynamic calibration shown in Table 1 is a concern for real applications and the method's generalizability beyond badminton analyses, although it is well considered. If this part is performed in a learning-based manner, it would be a great contribution in machine learning venues.

**Questions:**

Did incorporating stroke types as a feature contribute to the overall performance? It might be redundant after inputting the gesture.

---

> ### Author Response · Authors · 2025-11-21
> **Thanks for your comments**
>
> **Response to Weakness 1:** We thank the reviewer for raising this important point. The core novelty of our work lies in proposing a method to solve a **new and challenging problem of pre-stroke landing prediction**. This novel problem represents a fundamental shift from inferring the landing point from the ball's trace to predicting it from pre-stroke contextual cues. This shift enables significantly **shorter inference time** shown in Table 1, making it particularly suitable for robotic interception planning where rapid decision-making is crucial. In addition, by utilizing only pre-stroke information such as player positioning, posture, and shuttlecock approach patterns, our framework provides valuable insights for coaching and error analysis by revealing how these preparatory factors influence landing outcomes.
>
> While LSTMs and gated fusion are established techniques, we remark some differences in our work.
>
> **(1) Dynamic Spatio-temporal Decomposition:** Unlike traditional LSTM or gated fusion[a], this module extracts features from different domains (the spatial and the temporal) and merges them dynamically. Specifically, the spatial and temporal representations are extracted from the 3D trajectory of the shuttlecock through Transformer and CNN. These two modalities are merged by a learnable gate fusion dynamically.
>
> **(2) Conditional Cross Stage Fusion:** Unlike a standard LSTM, our mechanism dynamically prioritizes modalities based on the real-time inferred stroke context, a necessity due to the vastly different information content before a stroke is executed. This proposed gating approach integrates features across multiple processing stages vertically and horizontally, preserving original information while enabling deep feature interaction.
>
> **Table 1: Inference Time Analysis on TrackNetV2**
>
> | Component | Latency (ms) |
> |-----------|-------------|
> | Feature extraction | 127 |
> | ConFu prediction | 97 |
> | **Total end-to-end latency** | **224** |
> | Monotrack | 1478 |
> | ShuttleNet | 748 |
>
> **Response to weakness 2:** We sincerely thank the reviewer for this insightful suggestion. We fully agree that our initial aerodynamic-based method lacks generalizability, and the recommendation to employ learning-based approaches is highly valuable.
>
> Following this guidance, we have conducted extensive experiments evaluating **six distinct machine learning-based alternatives**:
>
> 1. **MLP**: A multi-layer perceptron for trajectory correction
> 2. **LSTM**: Sequence modeling of shuttlecock trajectory
> 3. **Transformer**: Attention-based trajectory refinement
> 4. **Random Forest**: Traditional ensemble method for error correction
> 5. **XGBoost**: Gradient boosting for trajectory optimization
> 6. **CNN-LSTM**: Hybrid architecture for spatiotemporal correction
>
> The experimental results (Table 2) demonstrate the following performance comparison:
>
> **Table 2: Performance Comparison of Different Recalibration Methods**
>
> | Method | MAE (m) | Domain Shift MAE (m) |
> |--------|---------|---------------------|
> | MLP | 0.234 | 0.309 |
> | LSTM | 0.208 | 0.282 |
> | Transformer | 0.218 | 0.289 |
> | Random Forest | 0.252 | 0.326 |
> | XGBoost | 0.243 | 0.319 |
> | CNN-LSTM | 0.204 | 0.266 |
> | **Ours (Physics-informed)** | **0.201** | **0.223** |
>
> **Response to Question:** Thank you for raising this question. While pre-stroke gesture is crucial, we think the stroke type is not totally redundant based on the understanding of the badminton sport. **The same pre-stroke movements can signal different intents and lead to different stoke type.** For instance, a player moving backward with a raised arm could be setting up for a clear or a smash.
>
> To assess whether stroke type provides non-redundant information in **real data**, we trained a multi-class classifier (ResNet-18 + TCN(Temporal Convolutional Network)) that predicts 10 stroke types using pre-stroke gesture data. The model achieved an accuracy of 82.5%, which suggests that the 82.5% stroke type information is redundant. And the remaining 17.5% misclassification rate indicates that stroke type contributes additional unique information.
>
> To evaluate the contribution of the stroke type feature to **our model**, we performed an ablation study in which this feature was removed when predicting the drop-point position. As shown in Table 3, excluding stroke type reduces the prediction accuracy by 4.2% and increases the MAE by 0.05 m, demonstrating its importance.
>
> **Table 3: Ablation Study on Stroke Type Feature Importance**
>
> | Model Configuration | MAE (m) | Accuracy (%) | Performance Drop |
> |---------------------|---------|--------------|------------------|
> | Full Model (with Stroke Type) | **0.20** | **92.6** | -- |
> | Without Stroke Type Feature | 0.25 | 88.4 | ↓ 25.0% / ↓ 4.2pp |

---

> ### Comment · Reviewer_5AJL · 2025-11-28
> **Post-rebuttal comment**
>
> I appreciate the authors' response.
> Most of my concerns are answered, and the results of machine-learning-based recalibration look insightful, showing the superiority of the physics-informed one so far.
> The remaining concern is the fitness of the venue, but I will update the score toward acceptance.

---

> > ### Author Response · Authors · 2025-11-30
> > **Thank you for your positive feedback and for updating your score towards **acceptance****
> >
> > Thank you for your positive feedback and for updating your score towards **acceptance**.
> >
> > We appreciate your time and valuable comments throughout the review process.

---

### Meta-Review · Area_Chair_JVmJ · 2026-01-08

**Summary:**

The main concerns were (1) that the paper is very specific and there was just one dataset comparison, (2) a lack of a fair comparison with existing work, and (3) that there was no comparison with learnable approaches. All the additional information that the authors provided should be reviewed again since the page limit has been exceeded and the valuable comments from the reviewers will improve the paper. Therefore, the decision is to reject the paper because all the results need to be re-evaluated.

**Reviewer Concerns:**

Most of the concerns were addressed. As mentioned before, the limited results were the main issue, and the author tried to tackle the issue. Therefore, most of them were not convinced due to the limited scope of the paper, even for the most positive reviewer.

**Reviewer Scores:**

Reviewers' opinions were included during the rebuttal, and they were likely to increase slightly the score from 0 to 2 and from weak rejection to weak acceptance. However, the overall opinion from the reviewers is the rejection of the paper because the paper were very specific and application-oriented, and also the most critical reviewer highlighted the lack of clarification compared to existing works, which is missing and probably wrong.

---

### Decision · Program_Chairs · 2026-01-26

Reject